# Genomic and phenotypic evolution of *Escherichia coli* in a novel citrate-only resource environment

**Zachary D Blount[1,2†]\*, Rohan Maddamsetti[3†]\*, Nkrumah A Grant[1,2†]\*, Sumaya T Ahmed[4], Tanush Jagdish[2,5], Jessica A Baxter[1], Brooke A Sommerfeld[1], Alice Tillman[4], Jeremy Moore[4], Joan L Slonczewski[4], Jeffrey E Barrick[2,6], Richard E Lenski[1,2]**

[1]Department of Microbiology and Molecular Genetics, Michigan State University, East Lansing, United States; [2]The BEACON Center for the Study of Evolution in Action, East Lansing, United States; [3]Department of Biomedical Engineering, Duke University, Durham, United States; [4]Department of Biology, Kenyon College, Gambier, United States; [5]Program for Systems, Synthetic, and Quantitative Biology, Harvard University, Cambridge, United States; [6]Department of Molecular Biosciences, The University of Texas, Austin, United States

**\*For correspondence:**
zachary.david.blount@gmail.com (ZDB);
rohan.maddamsetti@gmail.com (RM);
nkrumah.grant88@gmail.com (NAG)

[†]These authors contributed equally to this work

**Abstract** Evolutionary innovations allow populations to colonize new ecological niches. We previously reported that aerobic growth on citrate (Cit[+]) evolved in an *Escherichia coli* population during adaptation to a minimal glucose medium containing citrate (DM25). Cit[+] variants can also grow in citrate-only medium (DM0), a novel environment for *E. coli*. To study adaptation to this niche, we founded two sets of Cit[+] populations and evolved them for 2500 generations in DM0 or DM25. The evolved lineages acquired numerous parallel mutations, many mediated by transposable elements. Several also evolved amplifications of regions containing the *maeA* gene. Unexpectedly, some evolved populations and clones show apparent declines in fitness. We also found evidence of substantial cell death in Cit[+] clones. Our results thus demonstrate rapid trait refinement and adaptation to the new citrate niche, while also suggesting a recalcitrant mismatch between *E. coli* physiology and growth on citrate.

## Introduction

Evolutionary novelties are qualitatively new traits that allow populations to invade previously inaccessible ecological niches (*Simpson, 1953*; *Mayr, 1960*). Novel traits are thus important drivers of speciation and adaptive radiations that promote biodiversity and ecological complexity. Indeed, many major transitions in evolution have been mediated by novel traits such as photosynthesis, multicellularity, endoskeletons, sociality, and cognition (*Maynard Smith and Szathmary, 1997*, *Lundgren et al., 2016*, *Erwin, 2017*; *Erwin, 2019*).

We previously proposed a model in which novel traits can evolve in three distinct phases (*Blount et al., 2012*). In the potentiation phase, mutations accumulate in a lineage that make it possible to evolve the trait. In the actualization phase, a specific mutation produces the trait. Newly evolved traits are typically weak and ineffective. However, if the new trait confers even a slight advantage, it may spread throughout a population and, in the refinement phase, be improved by natural selection acting on subsequent mutations.

While potentiation and actualization enable the emergence of a novel trait, the capacity for refinement affects the trait's long-term persistence and potential to influence subsequent evolution (*Quandt et al., 2015*; *Erwin, 2015*). Prospects for refinement depend on the capacity to generate

heritable phenotypic variation that can improve the trait and integrate it with other aspects of organismal performance (*Kirschner and Gerhart, 1998*; *Pigliucci, 2008*), a facet of evolvability that we call 'refinement potential'.

Refinement potential is likely crucial for a population's long-term success in a new niche. A novel trait might allow a lineage to discover a new niche, but it does not guarantee long-term persistence. The new conditions may expose the population to selection pressures that differ in important respects from those of its ancestral niche, resulting in a mismatch between the organism and its environment (*Yeh, 2004*; *Schluter and Conte, 2009*; *Hu et al., 2017*). Successful establishment can depend on ameliorating this mismatch (*Chang et al., 2011*; *Turkarslan et al., 2011*), and failure to further adapt may lead to invasion failures (*Zenni and Nuñez, 2013*). Adaptation to a new niche therefore reflects a tension between evolvability and robustness (*Lenski et al., 2006*). The benefits of refining a novel trait must outweigh the costs (if any) of integrating that trait into organismal physiology.

Adaptation to novel niches has been widely studied in the context of invasive species that colonize and adapt to unfamiliar environments (*Davis, 2009*; *MacDougall et al., 2009*; *Logan et al., 2019*). However, the ongoing refinement of traits that provide access to novel niches has received little attention, probably because most evolutionary novelties (and associated niche discoveries) occurred in the distant past and are therefore difficult to study. Experimental evolution allows researchers to overcome this challenge. It is possible to study evolutionary novelties that arise during experiments with microbial (*Blount et al., 2008*; *Ratcliff et al., 2012*; *Barrick and Lenski, 2013*; *Kassen, 2019*) and digital (*Lenski et al., 2003*) systems, in which evolution can be studied in real-time.

One such system is the Long-Term Evolution Experiment with *Escherichia coli* (LTEE), in which 12 bacterial populations founded from a common ancestral strain have been propagated for >70,000 generations in a glucose-limited minimal medium, DM25 (*Lenski et al., 1991*). DM25 also contains abundant citrate, which serves as an iron-chelating agent (*Blount, 2016*). Many bacteria can grow aerobically on citrate, but most *E. coli* strains cannot because they are unable to transport citrate into the cell (*Koser, 1924*; *Hall, 1982*; *Reynolds and Silver, 1983*; *Pos et al., 1998*).

Citrate was unexploited as a carbon and energy source in all of the LTEE populations until a Cit$^+$ variant evolved in the population designated Ara−3 after ~31,000 generations (*Blount et al., 2008*). The Cit$^+$ trait arose in one of three coexisting lineages in this population by a genetic duplication that activated a previously unexpressed di- and tricarboxylate transporter (*Blount et al., 2012*). The benefit of this duplication mutation was contingent, at least in part, on that lineage's prior evolution of an enhanced ability to use acetate excreted into the medium as a byproduct of glucose metabolism. That enhanced ability resulted from a mutation in citrate synthase that altered carbon flow into the tricarboxylic acid cycle in a manner that was pre-adaptive for growth on citrate (*Quandt et al., 2015*). Concurrently, the supply of competing beneficial mutations of large effect declined over time in the LTEE, allowing the Cit$^+$ lineage to escape competitive exclusion (*Leon et al., 2018*). The Cit$^+$ trait radically altered this population's ecology and subsequent evolution (*Blount et al., 2008*; *Blount et al., 2012*; *Quandt et al., 2015*; *Quandt et al., 2014*; *Turner, 2015*; *Turner et al., 2015*). Access to the large citrate pool in the medium led to a several-fold increase in population size (*Blount et al., 2008*). Nonetheless, a Cit$^−$ lineage stably coexisted with the new Cit$^+$ lineage for some 10,000 generations, before finally going extinct (*Blount et al., 2008*; *Blount et al., 2012*; *Turner, 2015*; *Turner et al., 2015*). Even after 70,000 generations, none of the other 11 populations in the LTEE have evolved the ability to use the available citrate (*Blount et al., 2018*).

The emergence of Cit$^+$ in the LTEE provides a powerful model system for studying the process of evolutionary innovation. Cit$^+$ variants can grow not only in DM25, which contains both glucose and citrate, but also in DM0, a citrate-only medium in which *E. coli* normally cannot grow. How would the Cit$^+$ trait be refined if these variants colonized and adapted to this newly accessible citrate-only environment? To address this question, we founded 12 new, initially clonal Cit$^+$ populations and allowed them to evolve in DM0 for 2500 generations. We also allowed a second set of 12 populations to evolve in the original DM25 medium for comparison. We sequenced the genomes of evolved clones sampled from all 24 populations to find parallel genetic changes that indicate likely targets of selection (*Tenaillon et al., 2016*; *Deatherage et al., 2017*). Among other parallel changes, we identified numerous IS element insertions and several large gene amplifications. Our results thus show that genomic structural variation involving transposable elements and

amplifications can provide a rich source of plasticity and potential for novel trait refinement and adaptation to new niches.

We also compared the growth of the DM0-evolved clones to that of their ancestors in both DM0 and DM25, and we examined fitness changes at the level of both whole populations and individual clones. Although all populations show substantial adaptation reflected in their growth parameters, we also found evidence of persistent maladaptation, suggesting that this new function poses metabolic challenges that are difficult to overcome evolutionarily. Some individual evolved clones grow more poorly than their ancestors, even in the medium in which they had evolved. The fitness assays show atypically large variation across replicate assays of evolved populations and clones, as well as some paradoxical apparent declines in fitness despite 2500 generations of evolution. We also observed high levels of cell death in the ancestral and evolved Cit[+] clones that we examined. This experimental system thus sheds light not only on how new traits are refined during adaptation to a novel niche, but also on how maladaptive phenotypes may persist for long periods in new environments.

## Results

### Experimental design and phylogenetic analysis of sequenced strains

We isolated three Cit[+] clones (CZB151, CZB152, and CZB154) from the 33,000-generation sample of the Ara−3 population, and derived spontaneous Ara[+] revertants of each clone (ZDB67, ZDB68, and ZDB69, respectively). We used each of the six clones to found two populations that evolved in the citrate-only medium (DM0) for 2500 generations and two populations that evolved for 2500

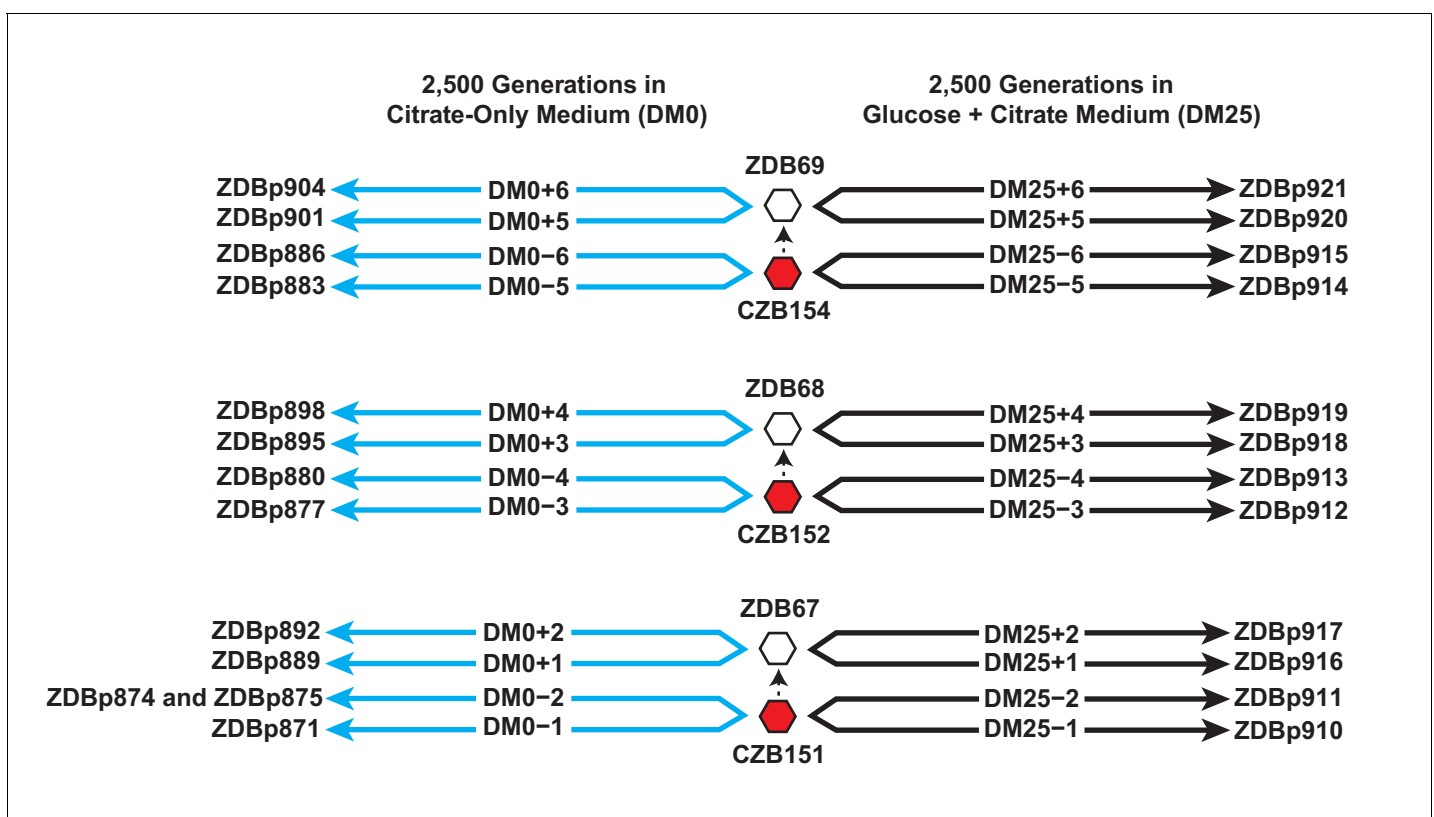

**Figure 1.** Experimental design and sequenced clone derivations. We isolated three Cit[+] clones (red hexagons) from generation 33,000 of LTEE population Ara−3. We then derived Ara[+] mutants (white hexagons) from those three LTEE clones. We used these six clones to found 24 populations. Twelve populations evolved for 2500 generations in citrate-only medium, DM0 (cyan lines). The remaining 12 evolved for 2500 generations in glucose and citrate medium, DM25 (black lines). The evolved clones we isolated after 2500 generations for genomic and phenotypic analysis are shown for each population.

generations in the medium containing both glucose and citrate (DM25) as a control (*Figure 1*). Evolved clones were isolated from each of the 24 populations at the end of the experiment, and we sequenced their genomes along with those of the six founding clones. We used these data to identify mutations that had accumulated during the evolution experiment.

We also used the genomic data to verify the presumed phylogenetic relationships among the ancestral (including the Ara+ revertants) and evolved clones, in the context of the Cit+ lineage of the Ara−3 population. This analysis showed that CZB154 is one mutation off the line of descent for the post-33,000 generation Cit+ lineage in the Ara−3 population, as it subsequently evolved in the LTEE (*Blount et al., 2012*). That mutation is a 1 bp deletion (GGGGGG → GGGGG) in the promoter of the hypothetical protein-coding gene ECB_03525. CZB151 does not have that mutation, but it possesses all of the other mutations found in the CZB154 clone, as well as two additional mutations. One is a C→G transversion that causes a nonsynonymous E181K mutation in the *insD* transposase. The other is a deletion of a CGCGG repeat that restores both the reading frame and function to the pseudogene *dcuS* (*Turner, 2015*). The restored gene encodes a histidine kinase that regulates anaerobic fumarate respiration (*The UniProt Consortium, 2017*; *Jeske et al., 2019*). CZB152, by contrast, belongs to a lineage somewhat farther from the eventual line of Cit+ descent in the Ara−3 population, and it differs from CZB151 and CZB154 by several mutations (*Blount et al., 2012*). Genomic analysis also showed that the Ara+ revertant ZDB67 differs from its parent clone, CZB151, only in the expected restoration-of-function mutation in the *araA* gene. The Ara+ revertants ZDB68 and ZDB69 have secondary mutations relative to CZB152 and CZB154, respectively, in addition to the expected mutation in *araA*. ZDB68 has a C→G transversion that introduces a nonsynonymous T33I mutation in *yfcC*, which encodes a predicted inner-membrane protein; and ZDB69 has a 1 bp deletion in *nplI,* which encodes a hypothetical protein of unknown function. All 24 evolved clones evolved in DM0 or DM25 have the mutations that are unique to their ancestors. Therefore, no cross-contamination that would compromise the independence of the evolved lines took place during the experiment.

One evolved clone from population DM0−2, ZDBp874, lacks the *citT* amplification that confers the Cit+ trait (*Blount et al., 2008*; *Blount et al., 2012*). That clone also displays a negative reaction on Christensen's Citrate agar, confirming a Cit− phenotype. We therefore sequenced the genome of a second isolate, ZDBp875, from the same population. We verified that ZDBp875 has the Cit+ trait. The ZDBp874 and ZDBp875 genomes share only a single derived mutation, an IS*150* insertion at the −35 position of the promoter of *yhiO*, which encodes universal stress protein B (UspB). The two clones thus appear to belong to coexisting lineages that diverged early during their evolution in DM0. We did not discover any additional Cit− variants in the DM0−2 population during a phenotypic screen of several hundred clones. Previous work has shown that the *citT* amplifications are prone to spontaneous collapse back to a single copy (*Blount et al., 2012*). This collapse, which presumably occurs by homologous recombination, eliminates CitT expression and causes reversion to the Cit− phenotype. The ZDBp874 clone could be either a recent and fortuitously sampled 'amplification collapse' mutant or a representative of a rare and stably coexisting Cit− lineage in that population.

## Genome evolution is faster in the citrate-only environment than in the control environment

The populations that evolved in the citrate-only DM0 medium accumulated more mutations than those that evolved in DM25, which contains both glucose and citrate (*Figure 2A* vs. *Figure 2B*). The DM0-evolved genomes had an average of 19.5 mutations, whereas the DM25-evolved genomes had an average of 13.7 mutations (Mann-Whitney two-tailed test, p=0.0116). The DM0 genomes had an average of 3.1 nonsynonymous SNPs in protein-coding genes, as compared to 1.1 on average for the DM25 genomes. The DM0 genomes also had more IS insertions on average than the DM25 genomes (10.3 vs. 6.6), driven largely by IS*150* insertions (8.5 vs. 4.9). The evolved genomes from the DM0 and DM25 treatments had similarly low average numbers of synonymous mutations (0.1 vs. 0.3), deletions (3.8 vs. 3.8), non-IS insertions (0.8 vs. 0.8), consecutive bp-substitutions (0.1 vs. 0.1), and SNPs outside of coding regions (1.4 vs. 1.1). The nearly identical numbers of synonymous mutations and SNPs outside protein-coding genes that we see in genomes evolved in DM0 and DM25 imply that the disparities in nonsynonymous mutations and IS insertions between the two conditions were, at least in part, driven by stronger selection in DM0, as opposed to a higher mutation rate or

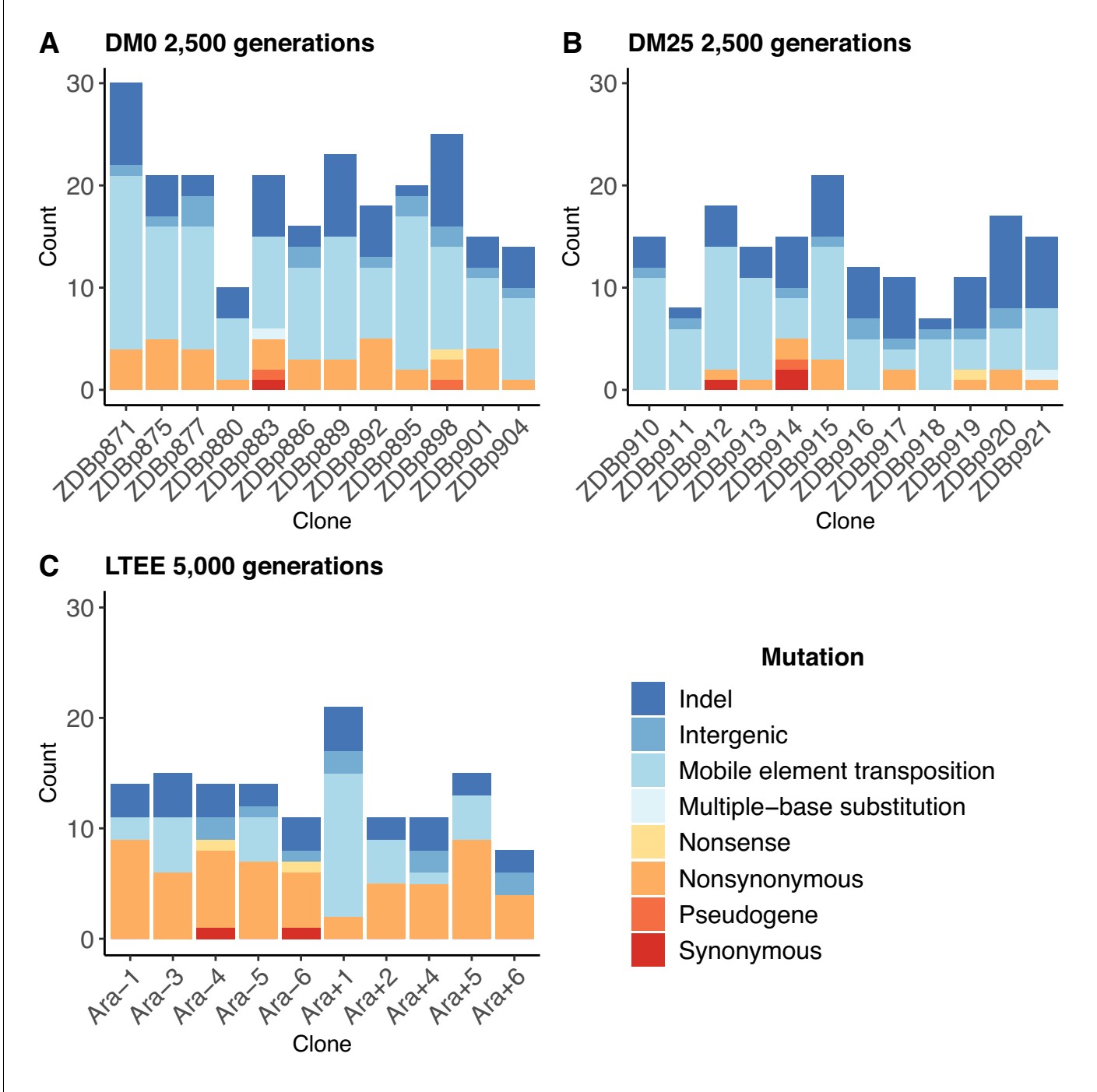

**Figure 2.** Numbers and types of mutations in evolved genomes. (**A**) Evolved genomes from the DM0 treatment after 2500 generations. (**B**) Evolved genomes from the DM25 treatment after 2500 generations. (**C**) Evolved genomes in the 10 non-hypermutable LTEE populations after 5000 generations. Mutations are color-coded according to the key: indel, insertions and deletions (excluding large duplications and amplifications); intergenic, intergenic point mutations; mobile-element transpositions; multiple-base substitution, consecutive point mutations (including adjacent to and in conjunction with indels); nonsense, nonsynonymous, and synonymous point mutations in protein-coding genes; pseudogene, mutations in pseudogenes.

The online version of this article includes the following source data for figure 2:

**Source data 1.** All evolved mutations found in the DM0-treatment and DM25-treatment clones.

**Source data 2.** Classification and counts of mutations in the 264 LTEE genomes, originally published as Supplementary Table 4 of *Tenaillon et al., 2016*.

differences in population dynamics caused by the more stressful DM0 environment (*Frenoy and Bonhoeffer, 2018*).

In both resource environments, the spectrum of mutations identified in evolved clones was dominated by structural variation, including insertions, deletions, and mobile element transpositions (*Figure 2A,B*). This spectrum is quite different from that observed in the LTEE populations. *Figure 2C* shows the number and spectrum of mutations in clones isolated from 10 LTEE populations after 5000 generations (two other populations had evolved point-mutation hypermutability and are not shown). Despite having evolved for twice as many generations, the LTEE clones have roughly similar numbers of mutations as observed in our experiments. The mutational spectrum was dominated by nonsynonymous point mutations in all but one of the LTEE populations, Ara+1. The mutation spectrum in our study is similar to that particular population, which evolved an elevated rate of IS*150* transposition early in its history (*Papadopoulos et al., 1999*; *Tenaillon et al., 2016*). It is also similar to that of a sub-lineage within another LTEE population, Ara−5, which also evolved IS*150*-mediated hypermutability, but much later in that experiment (*Tenaillon et al., 2016*). Most clones from the DM0 and DM25 treatments also have more deletions than the LTEE clones, again despite having evolved for fewer generations. These differences suggest some genomic instability in our study populations, in addition to the high rates of IS*150* transposition.

## Fitness changes after 2500 generations in DM0 and DM25 environments

We conducted competition assays with evolved population samples to measure their fitness in DM0 and DM25 (*Lenski et al., 1991*). We had difficulty in obtaining neutral derivatives of ancestral clones with the opposite Ara marker state, possibly due to genomic instability. We therefore used CZB151 as a common competitor for the Ara⁺ populations and ZDB67 for the Ara⁻ populations. Regardless of the environment in which they evolved, the populations display high variance across replicates in DM0 (*Figure 3A*), and most exhibit high variance across replicates in DM25 (*Figure 3B*). Eight of the 12 DM0-evolved populations have average fitness values in DM0 higher than their respective ancestral controls, but owing to the high variances, only two cases (DM0–6, DM0+6) appear compelling. Even in DM25, where the variances are less extreme, only two DM25-evolved populations (DM25–5, DM25–6) appear a bit more fit than their ancestors, while some DM0-evolved populations (DM0–1, DM0–2, DM0–5) were clearly less fit in DM25.

We also examined fitness changes in evolved clones relative to their direct ancestors. We only tested clones from populations for which we were able to obtain neutral ancestral variants with the opposite Ara marker state. We saw much lower variability across the clonal replicates than we did for the whole population samples. This reduced variance may reflect in part the higher replication and longer duration of the assays using clones; it might also be the case that the within-population genetic variation led to greater variation in the outcome of the whole-population competition assays. Nonetheless, we still saw inconsistent and paradoxical fitness changes in some clones. Two DM0-evolved clones (ZDBp880 and ZDBp886) were substantially less fit than their ancestors in DM0. All DM25-evolved clones except ZDBp913 were also less fit in DM0 (*Figure 4A*). In DM25, one DM0-evolved clone (ZDBp880) and one DM25-evolved clone (ZDBp915) were clearly less fit than their ancestors (*Figure 4B*).

## Changes in growth parameters after 2500 generations in DM0 and DM25 environments

To assess changes in growth parameters at the end of the evolution experiment, we compared the growth curves of evolved populations and clones to those of their respective ancestors (Figures 5–9). To quantify changes in growth parameters more precisely, we estimated the slope of the log-transformed growth curves over two separate intervals in DM25. In this medium, the Cit⁺ bacteria undergo an apparent diauxic shift from growth on glucose to growth on citrate. Therefore, we chose intervals of optical density (OD) in which the change in OD over time would correspond to the respective growth rates on those resources. We also estimated the duration of the lag prior to initial growth on glucose. A schematic of this method is shown in *Figure 5*. We calibrated the relevant intervals based on the growth kinetics of two Cit⁻ strains in DM25: the founding LTEE strain, REL606 (*Figure 5—figure supplement 1*), and the anomalous evolved clone, ZDBp874 (*Figure 7—figure*

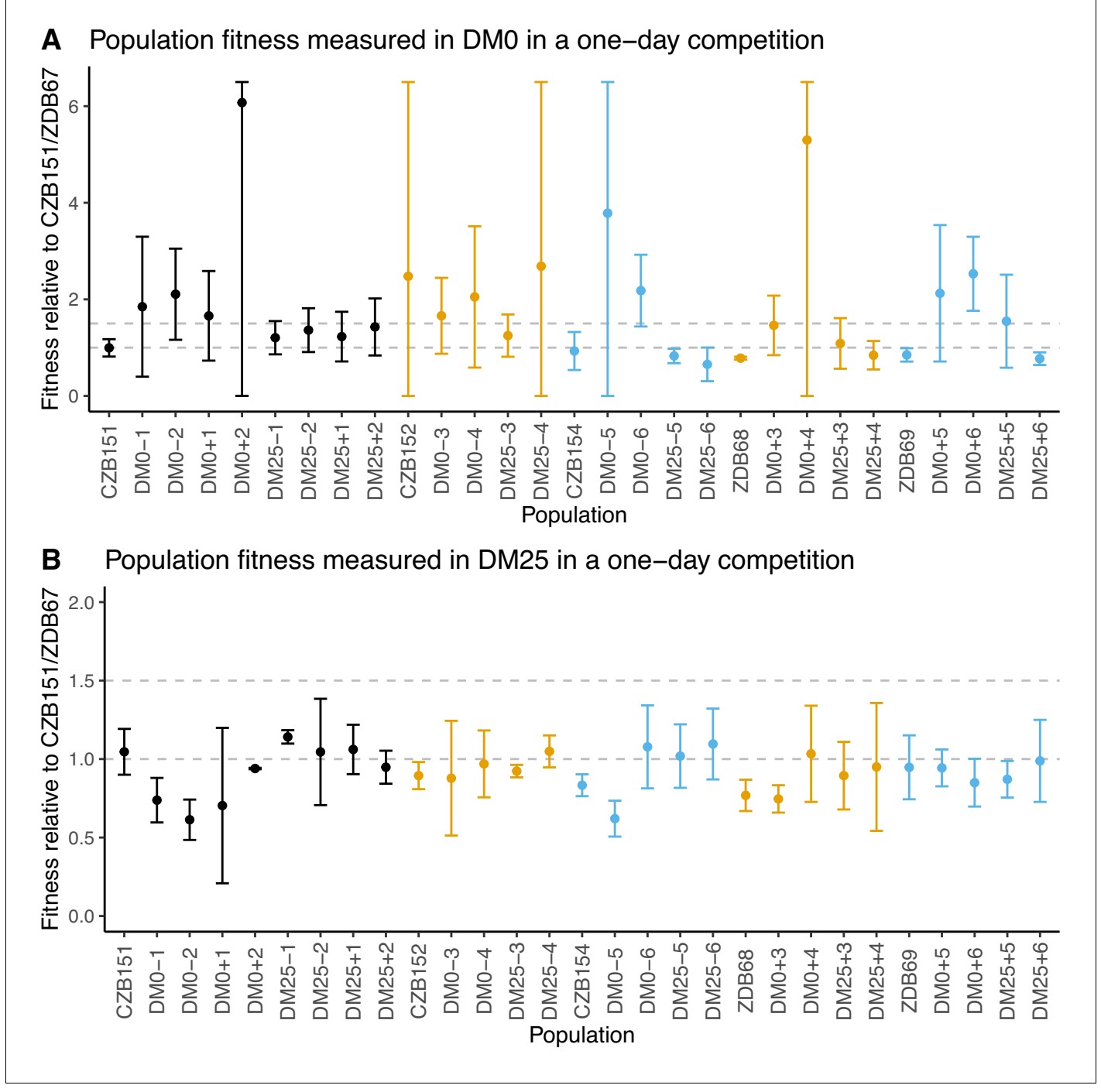

**Figure 3.** Fitness of evolved populations and their Cit[+] ancestors relative to Cit[+] ancestral clones CZB151 and ZDB67 in DM0 and DM25. To show the difference in scale across panels, dashed gray lines are drawn at 1.0 (neutrality) and 1.5 on the y-axis. Ancestral strain CZB151 and its descendants are shown in black, CZB152 and its descendants are in orange, and CZB154 and its descendants are in blue. (**A**) Fitness of evolved and ancestral populations relative to CZB151 and ZDB67 in DM0, as measured in one-day competition assays. Some confidence limits extend beyond the range shown on the y-axis. (**B**) One-day fitness of evolved and ancestral populations relative to CZB151 and ZDB67 in DM25, as measured in one-day competition assays. Error bars are 95% confidence intervals.

The online version of this article includes the following source data for figure 3:

**Source data 1.** Colony counts for fitness competitions of evolved populations in DM0 growth medium (Panel A).

**Source data 2.** Colony counts for fitness competitions of evolved populations in DM25 growth medium (Panel B).

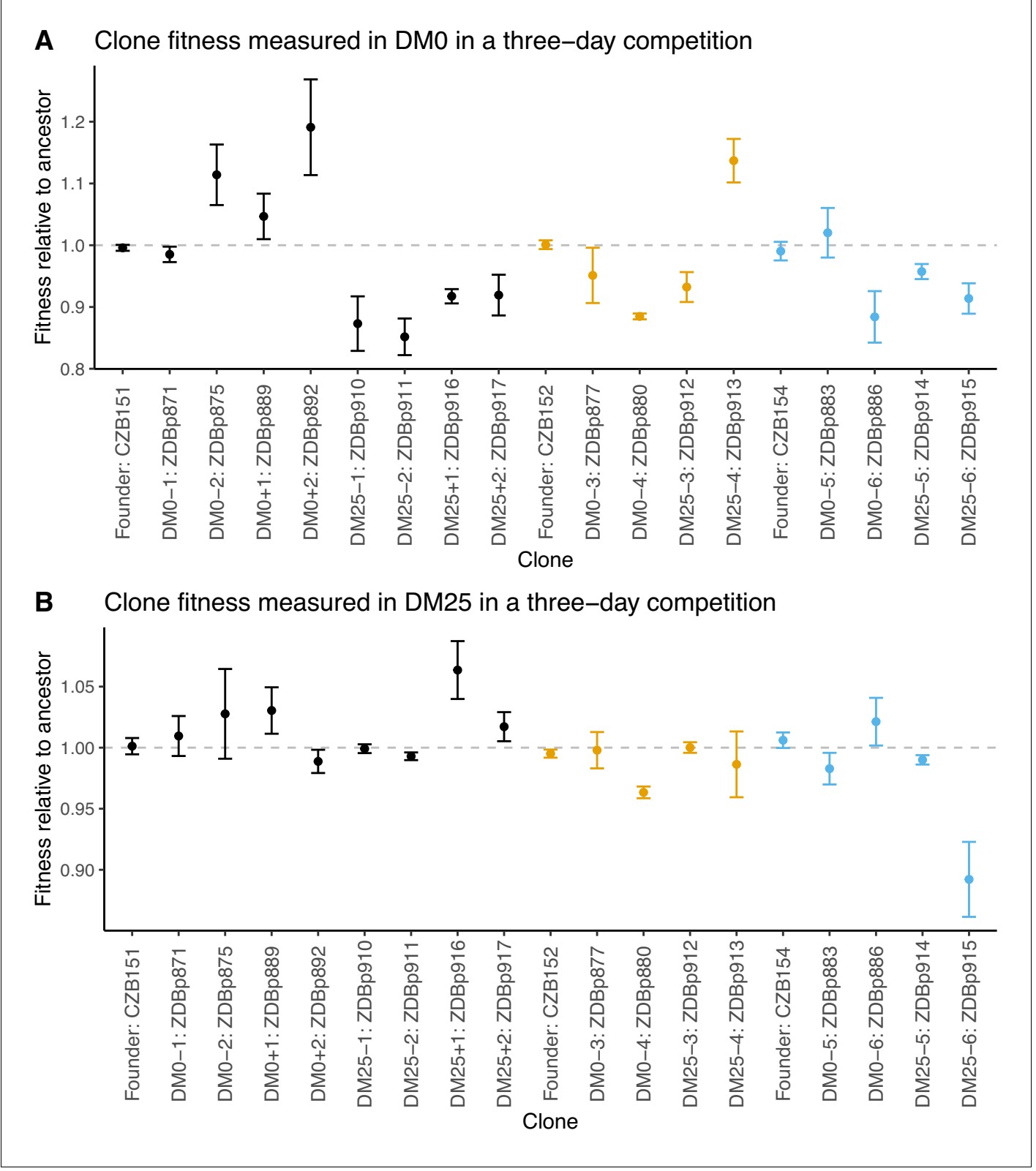

**Figure 4.** Fitness of select evolved clones against their direct ancestors in DM0 and DM25. The dashed grey line shows neutrality. Ancestral strain CZB151 and its descendants are shown in black, CZB152 and its descendants are in orange, and CZB154 and its descendants are in blue. (A) Fitness of evolved clones relative to their direct ancestors in DM0 in a three-day competition assay. (B) Fitness of evolved clones relative to their direct ancestors

*Figure 4 continued on next page*

*Figure 4 continued*

in DM25 in a three-day competition assay. Error bars are 95% confidence intervals. We selected clones for fitness assays based only on the availability of ancestral genotypes with confirmed, neutral, opposing Ara marker states.

The online version of this article includes the following source data for figure 4:

**Source data 1.** Colony counts for fitness competitions of evolved clones in DM0 growth medium (Panel A).
**Source data 2.** Colony counts for fitness competitions of evolved clones in DM25 growth medium (Panel B).

supplements *1* and *2*). In DM0, we estimated the duration of the lag phase and the growth rate on citrate only.

On balance, the populations evolved higher exponential growth rates and shorter lag phases in both DM0 and DM25. These demographic changes are consistent with those observed in the LTEE (*Vasi et al., 1994*). Indeed, all DM0-evolved populations show improvements in their growth on citrate in both DM0 and DM25 (*Figure 6*), whereas their growth on glucose in DM25 shows little or no change. In DM0, the populations also exhibit markedly reduced lags prior to commencing growth (*Figure 6—figure supplements 1* and *2*).

We observed substantially more variation in growth parameters among the evolved clones (*Figures 7* and *8*) than among the whole-population samples from which they were isolated. In fact, some evolved clones grow more poorly than their ancestors, as shown by non-overlapping confidence intervals on their growth parameter estimates in *Figure 7A*. Two CZB151-derived, DM0-evolved clones, ZDBp871 and ZDBp889, show little or no improvement in DM0, and they are markedly worse than CZB151 in DM25. Similarly, the anomalous Cit⁻ clone, ZDBp874, is not only unable to grow in DM0, but also grows much more poorly than its ancestor in DM25 (*Figure 7—figure supplements 1* and *2*). All other DM0-evolved clones grow better than their ancestors in DM0, and most also grow about as well as their ancestors in DM25, with the additional exception of ZDBp901.

Our finding that some evolved clones are significantly less fit than their ancestor, along with the differences in the growth parameters of the evolved clones and the whole populations, implies that ecologically relevant genetic variation exists in both the DM0- and the DM25-evolved populations. We therefore considered the possibility that improved growth performance on citrate always comes at a cost of reduced growth on glucose. We used our separate estimates of growth rates on glucose and citrate in DM25 to determine if growth on the two substrates was correlated. However, we found no significant correlation between growth rates on citrate and glucose for either the DM0-evolved clones or whole populations (*Figure 9A*). By contrast, the growth rates measured on citrate in the two media, DM0 and DM25, are highly correlated for both clones and populations (*Figure 9B*).

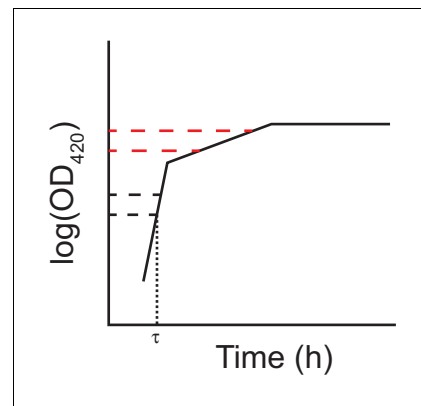

**Figure 5.** Schematic of the log-slope method to calculate growth rates. We log$_e$-transformed optical densities, and used the slope of the curve in the interval OD$_{420\ nm}$ = [0.01, 0.02] to calculate the exponential growth rate on glucose (h$^{-1}$), $r_{glucose}$. We used the slope of the curve in the interval OD$_{420\ nm}$ = [0.05, 0.1] to calculate the exponential growth rate on citrate (h$^{-1}$) $r_{citrate}$. In making this interpretation, we assumed a diauxic shift between growth on glucose and citrate, rather than simultaneous growth on both substrates. In any case, growth rates during these intervals are relevant phenotypes even without assuming diauxie. We estimated lag time ($\tau$) as the time (h) until OD$_{420\ nm}$ = 0.01 was reached.

The online version of this article includes the following source data and figure supplement(s) for figure 5:

**Figure supplement 1.** Growth curves for REL606 in DM25.

**Figure supplement 1—source data 1.** Optical density (420 nm) timeseries for REL606 growth in DM25 medium over more than 24 hr.

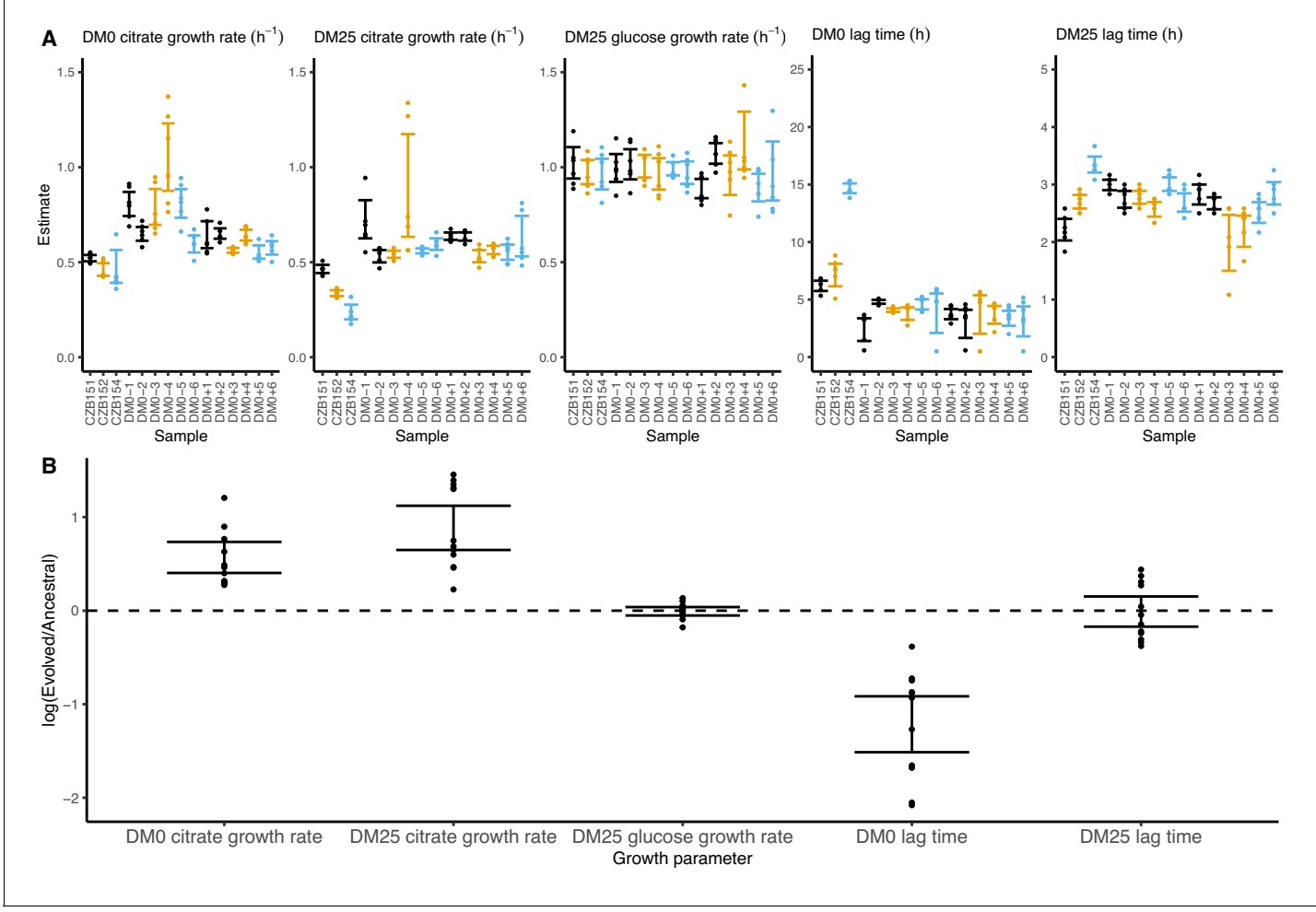

**Figure 6.** Growth parameters for whole-population samples that evolved in DM0 and their Cit⁺ ancestors. (**A**) Estimates of various growth parameters for the ancestral strains and DM0-evolved populations at 2500 generations, using the log-slope method. Ancestral strain CZB151 and its descendants are shown in black, CZB152 and its descendants are in orange, and CZB154 and its descendants are in blue. Units for growth rates are h$^{-1}$, and units for lag times are h. Bias-corrected and accelerated ($BC_a$) bootstrap 95% confidence intervals around parameter estimates were calculated using 10,000 bootstraps. (**B**) Estimates of log$_2$-transformed ratios of growth parameters for the evolved populations and their ancestors. The growth curves we used to estimate these parameters are shown in *Figure 6—figure supplements 1* and *2*.

The online version of this article includes the following source data and figure supplement(s) for figure 6:

**Source data 1.** Optical density (420 nm) timeseries for DM0-evolved populations and their ancestors in DM0 and DM25 growth media.

**Figure supplement 1.** Growth curves of the 12 DM0-evolved whole-population samples, measured in DM0 and DM25.

**Figure supplement 2.** Log$_e$-transformed growth curves of the 12 DM0-evolved whole-population samples, measured in DM0 and DM25.

## Evidence of cell death in clones isolated from both DM0 and DM25 environments

The contribution of cell death to fitness in the LTEE is generally negligible compared to that of growth (*Vasi et al., 1994*). However, we serendipitously discovered evidence of substantial cell death in cultures of a Cit⁺ clone sampled from the Ara−3 population of the LTEE at 50,000 generations. This observation led us to examine the relationship between the Cit⁺ trait and cell death in more detail by using fluorescence microscopy (*Figure 10*). We analyzed five clones: the LTEE ancestor, REL606; the 33,000-generation Cit⁺ clone, CZB151; one of its DM0-evolved descendants, ZDBp871; one of its DM25-evolved descendants, ZDBp910; and the 50,000-generation Cit⁺ clone, REL11364. We labeled cells from 24 hr stationary-phase cultures (i.e., when they would be transferred to fresh medium in the evolution experiment) using two-color live/dead stains (Materials and methods).

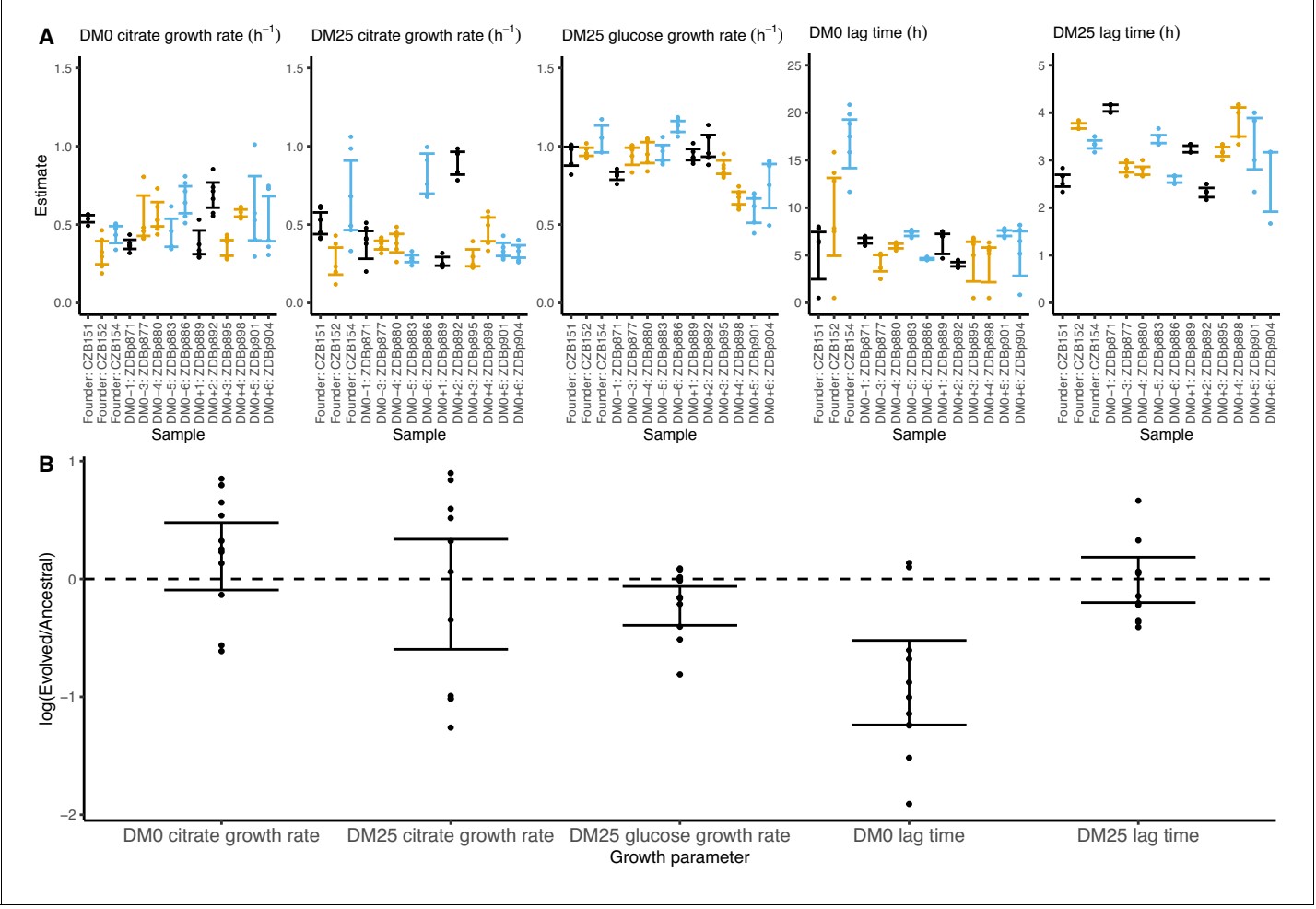

**Figure 7.** Growth parameters for clones from populations that evolved in DM0 and their Cit⁺ ancestors. (**A**) Estimates of growth parameters for the ancestral strains and DM0-evolved clones sampled at 2500 generations, using the log-slope method. CZB151 and its descendants are in black, CZB152 and its descendants are in orange, and CZB154 and its descendants are in blue. (**B**) Estimates of $\log_2$-transformed ratios of growth parameters for the evolved clones and their ancestors. The growth curves we used to estimate parameters are shown in *Figure 7—figure supplements 1* and *2*. We excluded the anomalous evolved Cit⁻ clone. See *Figure 6* for additional details.

The online version of this article includes the following source data and figure supplement(s) for figure 7:

**Source data 1.** Optical density (420 nm) timeseries for DM0-evolved clones and their ancestors in DM0 and DM25 growth media.

**Figure supplement 1.** Growth curves of the 12 DM0-evolved clones, measured in DM0 and DM25.

**Figure supplement 2.** $\log_e$-transformed growth curves of the 12 DM0-evolved clones, measured in DM0 and DM25.

Proportions of dead cells were calculated for five independent cultures for each clone and medium combination (except ZDBp910, for which we had problems with growth in DM0 and so have only one replicate, and REL606, which cannot grow in DM0). *Figure 10A* shows representative fields for each clone in DM0 and DM25. *Figure 10B* shows the resulting estimates of the proportion of dead cells, along with 95% bias-corrected and accelerated ($BC_a$) bootstrap confidence intervals (*DiCiccio and Efron, 1996*) weighted by the number of cells analyzed and scored in the replicate cultures. On average, 10.7% of the LTEE ancestral cells grown in DM25 were scored as dead in stationary phase. By contrast, when grown in the same DM25 medium, 29.6% and 39.9% of cells were scored as dead for the Cit⁺ clones isolated from LTEE population Ara−3 at 33,000 (CZB151) and 50,000 generations (REL11364), respectively. We observed similarly high proportions of dead cells for both clones in DM0 as well (33.1% and 44.2% for CZB151 and REL11364, respectively). These results indicate that the evolution of aerobic growth on citrate in the LTEE was associated with elevated mortality. Moreover, the increased mortality was not remedied even after almost 20,000

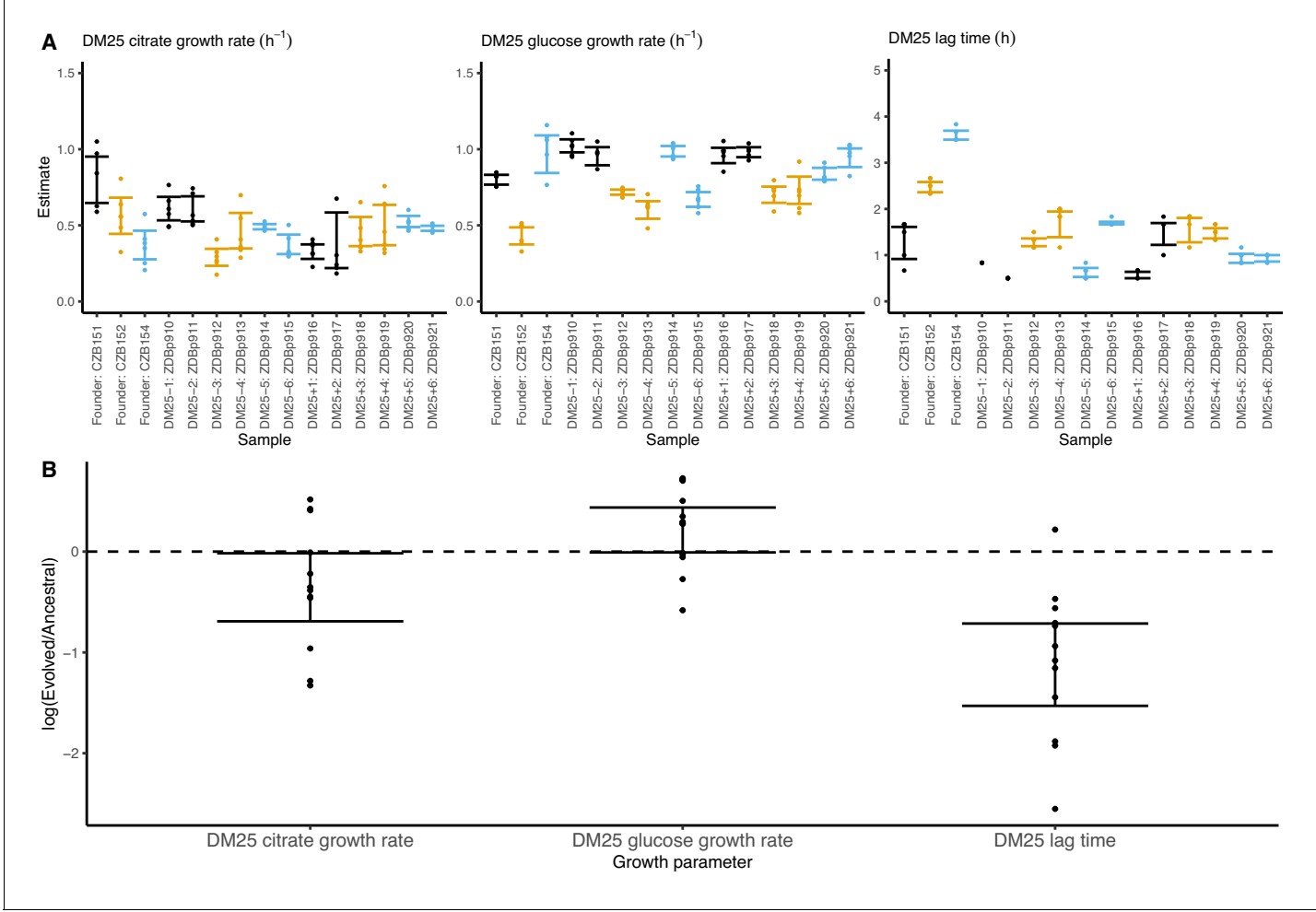

**Figure 8.** Growth parameters of the 12 DM25-evolved clones and their 3 Cit[+] ancestors. (**A**) Estimates of growth parameters for each ancestral and DM25-evolved clone, using the log-slope method (*Figure 2*). Estimates for ancestral strain CZB151 and its descendants are shown in black, estimates for CZB152 and its descendants are in orange, and estimates for CZB154 and its descendants are in blue. Units for growth rates *r* are h[−1], and units for lag times are h. Bias-corrected and accelerated (*BCa*) bootstrap 95% confidence intervals around parameter estimates were calculated using 10,000 bootstraps; no confidence interval is shown if a parameter could not be estimated accurately from the available data. Aberrant estimates that fall outside of these ranges are not shown. (**B**) Estimates of log2-transformed ratios of growth parameters for the evolved clones and their ancestors. The growth curves used to estimate these parameters are shown in *Figure 8—figure supplements 1* and *2*.

The online version of this article includes the following source data and figure supplement(s) for figure 8:

**Source data 1.** Optical density (420 nm) timeseries for DM25-evolved clones and their ancestors in DM25 growth medium.
**Figure supplement 1.** Growth curves of the 12 DM25-evolved clones, measured in DM25 only.
**Figure supplement 2.** Log_e-transformed growth curves of the 12 DM25-evolved clones, measured in DM25.

generations since the new trait arose in the Ara−3 population. The two evolved clones we examined from our evolution experiment, ZDBp871 and ZDBp910, show somewhat different patterns. Both show lower mortality in glucose-containing DM25 (25.3% and 12.4% for ZDBp871 and ZDBp910, respectively) but higher mortality in citrate-only DM0 (53.0% and 51.6% for ZDBp871 and ZDBp910, respectively). The reduced mortality of ZDBp910 in DM25, in which it evolved for an additional 2500 generations, suggests that the apparent metabolic imbalance associated with growth on citrate may be reduced by evolving in a medium that also contains glucose. It is even more surprising, then, that we observed no comparable reduction in mortality in the 50,000-generation Ara−3 clone, which might indicate that historically contingent ecological and genetic interactions are important for this trait. Moreover, the very high death rate of ZDBp871 in DM0, the medium in which it evolved,

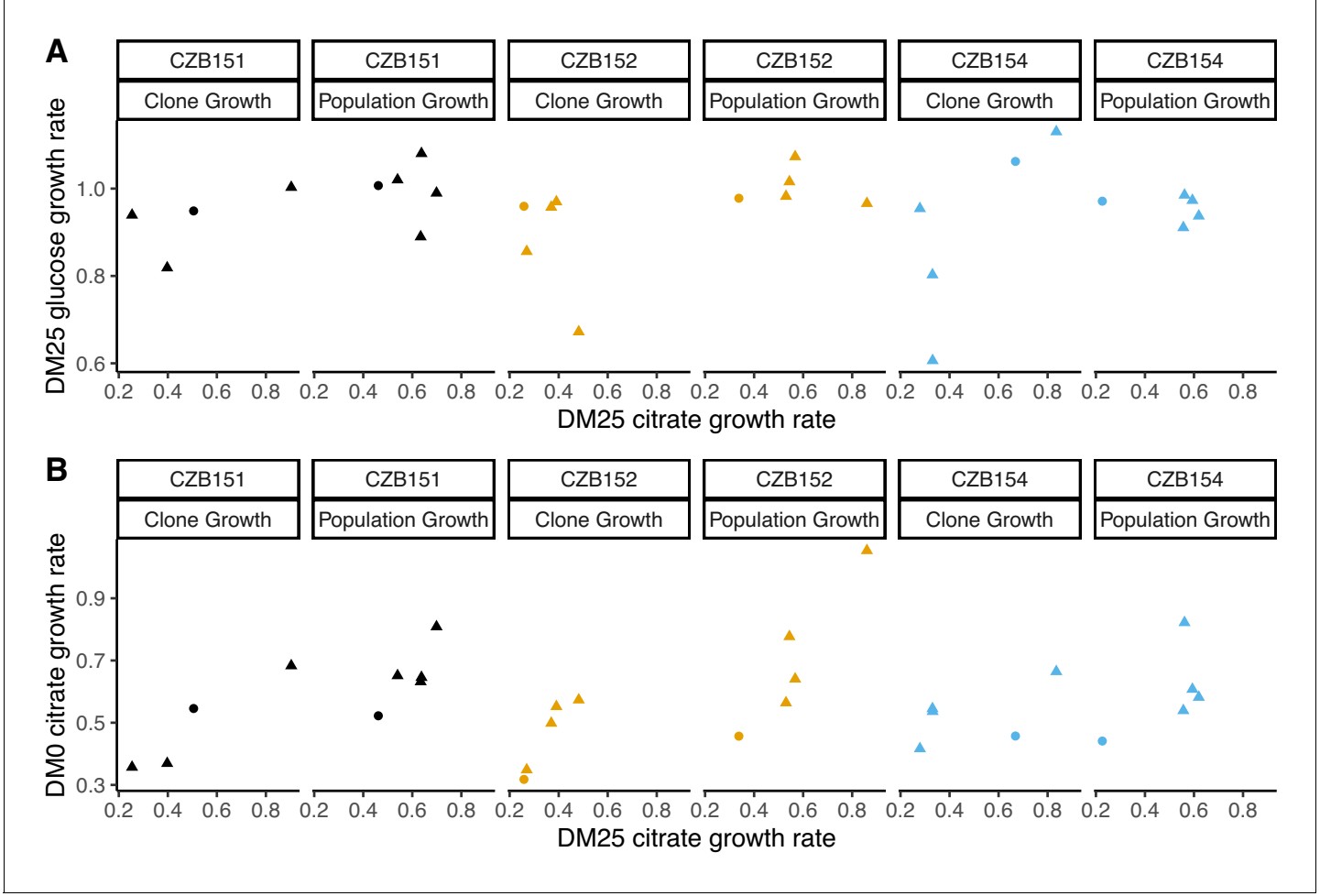

**Figure 9.** Correlations between estimated growth rates across substrates and media for DM0-evolved clones and populations. All tests are two-tailed, because growth rates across substrates and media might, in principle, exhibit tradeoffs. (**A**) Correlations between $r_{glucose}$ and $r_{citrate}$ in DM25 are not significant (Pearson's $r = 0.4788$, d.f. = 12, p=0.0833 for clones; $r = -0.0392$, d.f. = 13, p=0.8897 for populations). (**B**) Correlations between $r_{citrate}$ in DM0 and $r_{citrate}$ in DM25 are highly significant ($r = 0.7513$, d.f. = 12, p=0.0020 for clones; $r = 0.8041$, d.f. = 13, p=0.0003 for populations). Circles and triangles indicate ancestral and evolved samples, respectively. Colors distinguish the different Cit[+] ancestors and their evolved descendants.

The online version of this article includes the following source data for figure 9:

**Source data 1.** Optical density (420 nm) timeseries for DM0-evolved populations and their ancestors in DM0 and DM25 growth media.

**Source data 2.** Optical density (420 nm) timeseries for DM0-evolved clones and their ancestors in DM0 and DM25 growth media.

suggests that correcting the metabolic imbalance is even more difficult when citrate is the sole carbon and energy source.

## Specificity of genome evolution in the DM0 and DM25 environments

We found evidence that the DM0 and DM25 environments selected for mutations in different genes. Following *Deatherage et al., 2017*, we compared the distribution of 'qualifying' mutations—nonsynonymous SNPs, deletions, duplications, and IS insertions that unambiguously affect single genes— that arose during evolution in each medium. We identified all genes in which we found at least two qualifying mutations across the 24 evolved Cit[+] clones we sequenced. These genes are shown in *Figure 11*, where they are ranked by the absolute value of the difference in the number of qualifying mutations between the DM0 and DM25 conditions.

We then used the method of *Deatherage et al., 2017* to quantify the extent of parallelism in genome evolution within and between the DM0 and DM25 treatments. We computed Dice's Coefficient of Similarity, $S$, for each pair of evolved clones, where $S = 2 |X \cap Y|/(|X| + |Y|)$. $|X|$ and $|Y|$ are

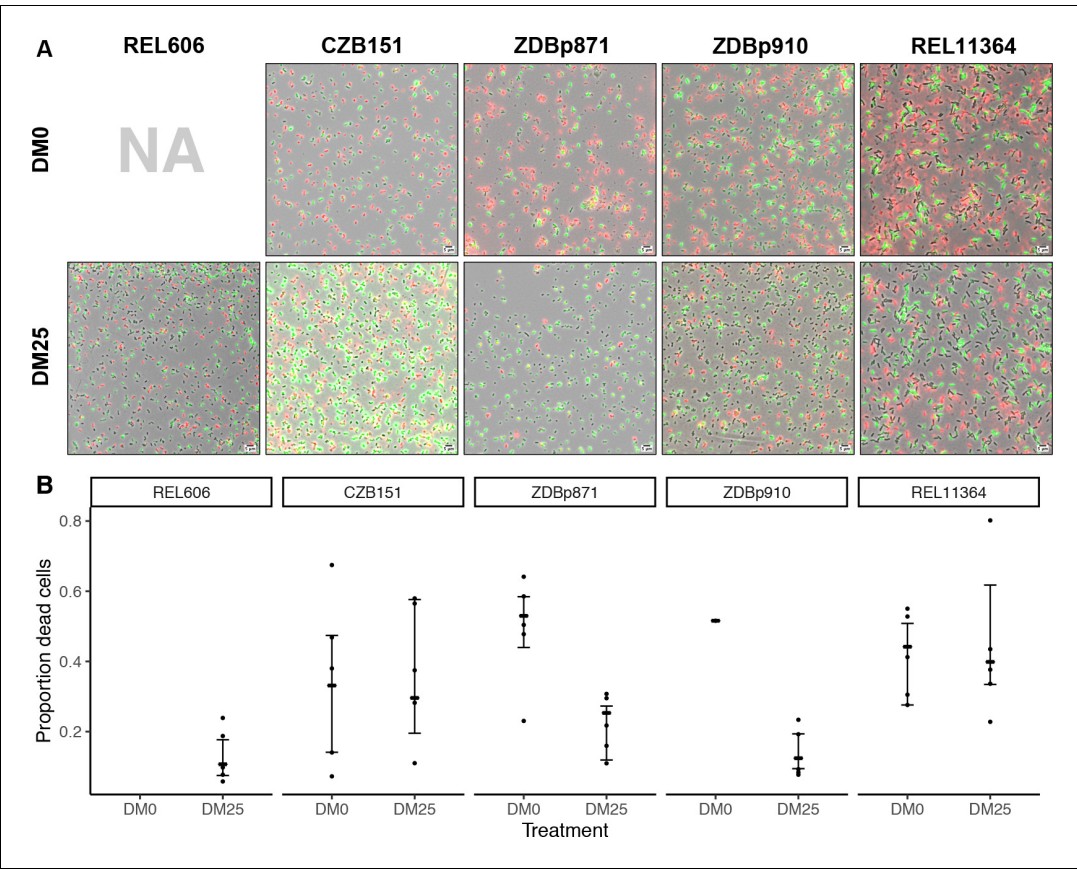

**Figure 10.** Elevated mortality in Cit$^+$ strains. The Cit$^+$ strains exhibit substantially elevated mortality in the citrate-only DM0 medium; some also show high mortality in DM25 as well. REL606 is Cit$^-$ and cannot grow in DM0. CZB151 was isolated from LTEE population Ara−3 at generation 33,000, and its descendants, ZDBp871 and ZDBp910, had evolved for 2500 generations in DM0 and DM25 media, respectively. REL11364 was isolated from LTEE population Ara−3 at generation 50,000. (**A**) Representative micrographs of the five clones in the two media. We stained cells using the *Bac*Light Viability Kit, and we scored them as dead if their red fluorescence exceeded their green fluorescence (see Materials and methods). Scale bars (lower right corner) represent 5 μm. (**B**) Proportion of dead cells in five replicate cultures of each strain grown in DM0 and DM25 medium each (except for ZDBp910, with only one replicate). The wider symbols show estimated overall proportions weighted by the number of cells analyzed in each replicate culture. We calculated bias-corrected and accelerated (*BC$_a$*) bootstrap 95% confidence intervals using 10,000 bootstraps (except for ZDBp910), and we weighted by the number of cells analyzed in each replicate.

The online version of this article includes the following source data for figure 10:

**Source data 1.** Cell death estimates.
**Source data 2.** Micrograph image segmentation and classification by SuperSegger software, part 1.
**Source data 3.** Micrograph image segmentation and classification by SuperSegger software, part 2.

the cardinalities of the sets of genes with qualifying mutations in two clones, and $|X \cap Y|$ is the cardinality of the set of genes with mutations in both clones. $S$ thus ranges from 0, when the two clones have no qualifying mutations in common, to 1, when both clones have qualifying mutations in exactly the same set of genes. The grand mean similarity, $S_m$, is 0.135 across the 24 evolved clones. The mean within-treatment similarity, $S_w$, is 0.177, meaning that two clones that evolved independently in the same medium on average have 17.7% of mutated genes in common. By contrast, the mean between-treatment similarity, $S_b$, is 0.096, meaning that two clones that evolved in different media on average have only 9.6% of mutated genes in common. We evaluated the significance of the difference between $S_w$ and $S_b$ using a randomization test in which clones were permuted across samples 10,000 times, and the difference between the two measures was calculated for each

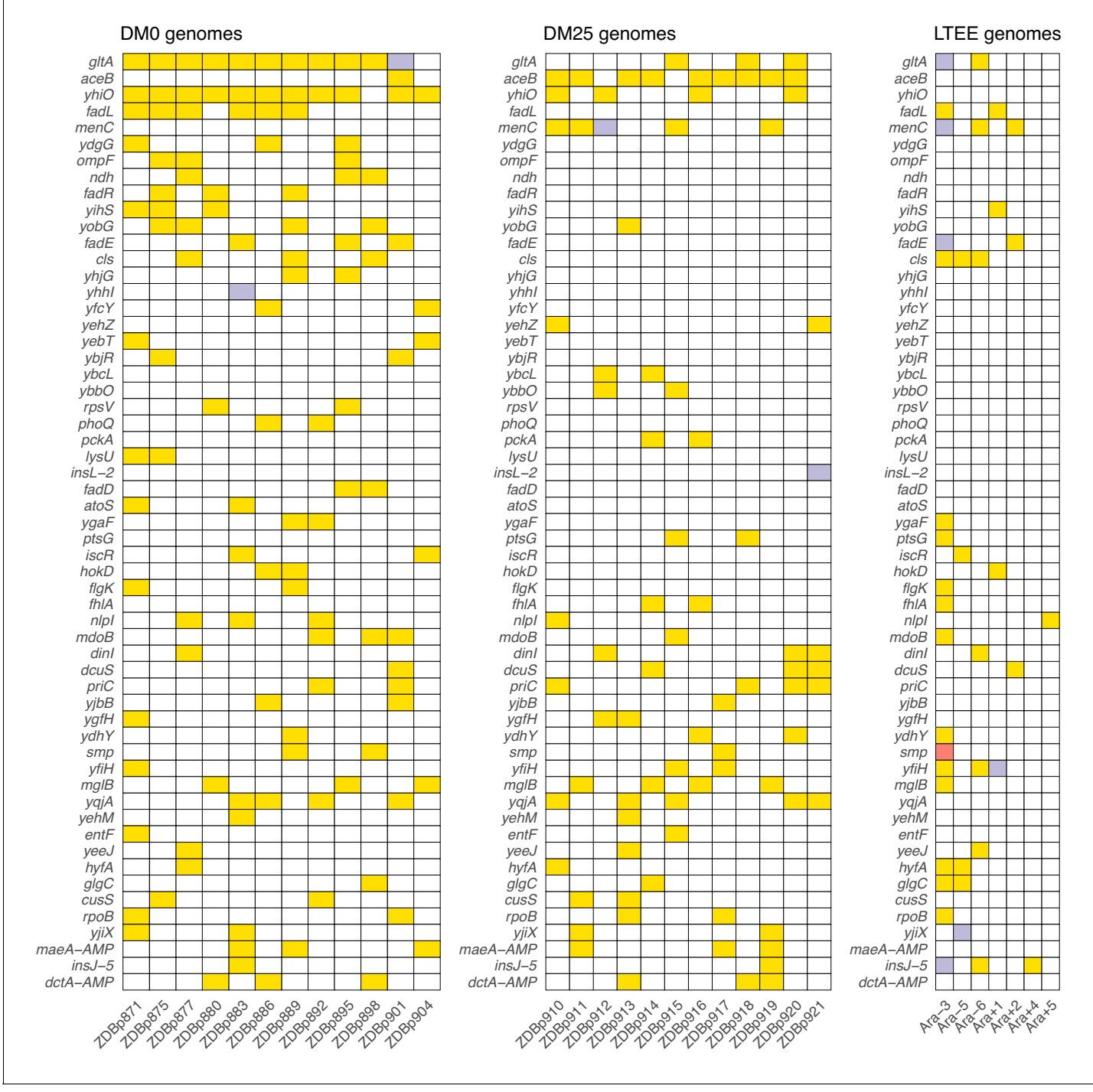

**Figure 11.** Parallel genetic evolution. Genes with mutations in two or more sequenced genomes from the DM0- and DM25-evolved populations, ranked by the absolute value of the difference in the number of qualifying mutations (see main text) between DM0 and DM25. Mutations in the same genes in the six non-mutator LTEE lineages and in a Cit$^+$ clone from LTEE population Ara−3 (which evolved hypermutability), all at 50,000 generations, are shown for comparison. Yellow, violet, or red fill indicates the presence of one, two, or three qualifying mutations, respectively. Mutation details are provided in ***Supplementary file 5***.

The online version of this article includes the following source data and figure supplement(s) for figure 11:

**Source data 1.** Counts of qualifying mutations in evolved clones.

**Source data 2.** Presence/absence of *dctA* and *maeA* amplifications in evolved clones.

**Source data 3.** Counts of qualifying mutations in non-mutator LTEE 50,000 generation clones.

**Figure supplement 1.** Parallel substitutions at the amino-acid level in citrate synthase, GltA.

*Figure 11 continued on next page*

*Figure 11 continued*

**Figure supplement 1—source data 1.** Annotation of evolved GltA residues on PDB structure 1NXG.

permutation. The observed difference between the DM0- and DM25-evolved clones was higher than in any of the permutations. The greater genomic parallelism within than between environments is therefore highly significant ($p < 0.0001$).

Five genes had significantly more parallel mutations in one environment than in the other. Eleven of the 12 DM0-evolved Cit$^+$ clones had qualifying mutations associated with *yhiO*, encoding the universal stress protein UspB, compared to 4 of 12 clones that evolved in DM25 (Fisher's exact test: p=0.0094). Similarly, we found qualifying mutations in *gltA*, which encodes citrate synthase, in 11 of the DM0-evolved Cit$^+$ clones, whereas only 3 of the DM25-evolved clones had mutations in that gene (Fisher's exact test: p=0.0028). The gene encoding isocitrate lyase, *aceB*, had only one qualifying mutation among the DM0-evolved genomes, but nine in the DM25-evolved genomes (Fisher's exact test: p=0.0028). Among the DM0-evolved genomes, there are no qualifying mutations in *menC*, which encodes O-succinylbenzoate synthase, but 5 DM25-evolved genomes have mutations in that gene (Fisher's exact test: p=0.0373). Six DM0-evolved genomes have qualifying mutations in the *fadL* gene, which encodes an outer membrane long-chain fatty acid channel, but none of the DM25-evolved genomes have mutations in this gene (Fisher's exact test: p=0.0137). Moreover, we found nine additional qualifying mutations associated with four other genes (*fadA*, *fadE*, *fadD*, and *fadR*) in the fatty-acid degradation regulon among the DM0-evolved genomes, but none in the DM25-evolved genomes. Mutations in the *fad* regulon thus show a strong signature of adaptation specific to the DM0 medium. Thirteen of the 15 qualifying mutations in the *fad* regulon were mobile-element insertions.

The environment was much more important than the ancestral genotype in determining the genetic targets of selection. We found no difference in the total number of qualifying mutations between the 24 evolved Cit$^+$ clones when grouped by ancestor (i.e., CZB151, CZB152, CZB154) (Kruskal-Wallis test, p=0.8873). Moreover, by using the same randomization test described above to test the significance of the difference between $S_w$ and $S_b$, we found no significant difference based on ancestral genotype (p=0.5540, based on 10,000 replicates).

We also found five instances of parallel changes at the amino-acid level among the DM0-evolved genomes. Three of the five occurred in *gltA*, which encodes citrate synthase: M172I, A162T, I114F. All three of these substitutions are near the allosteric binding pocket for NADH (*Figure 11—figure supplement 1*). *Quandt et al., 2015* reported an A162V substitution that likewise affects NADH binding, and which was previously shown to fine-tune carbon flux through citrate synthase (*Maurus et al., 2003*). These three *gltA* mutations presumably have similar effects. We also saw parallel I197L substitutions in *ygaF*, which encodes a protein that dehydrogenates L-2-hydroxyglutarate to alpha-ketoglutarate and replenishes the cell's reduction potential by feeding electrons from this reaction into the membrane quinone pool (*Kalliri et al., 2008*). There were parallel S351C substitutions in *atoS* in ZDBp871 and the anomalous Cit$^-$ clone ZDBp874. This gene encodes the sensor protein of a two-component regulatory system that stimulates short-chain fatty acid catabolism. Unlike some mutations that might reduce or destroy a protein's functionality, we expect that these parallel amino-acid substitutions fine-tune protein function (*Maddamsetti et al., 2017*).

## Contribution of transposable insertion elements to parallel evolution

Notwithstanding the parallel amino-acid substitutions described above, most of the parallel genomic evolution reflects the activity of IS elements. In both environments, most new IS insertions are copies of IS*150* elements (*Figure 12A and B*). We compared the number of IS*150* insertions in clones evolved in the two media to the number that had accumulated through 50,000 generations in the Ara−3 population of the LTEE (*Figure 12C*). The rates of IS*150* insertion accumulation in the Ara−3 population and the DM25-evolved Cit$^+$ populations are comparable, but much lower than in the DM0-evolved populations. The difference between the DM0- and DM25-evolved genomes is significant (Mann–Whitney *U* test, two-tailed p=0.0089), despite the high variability between genomes within each group (*Figure 2A and B*).

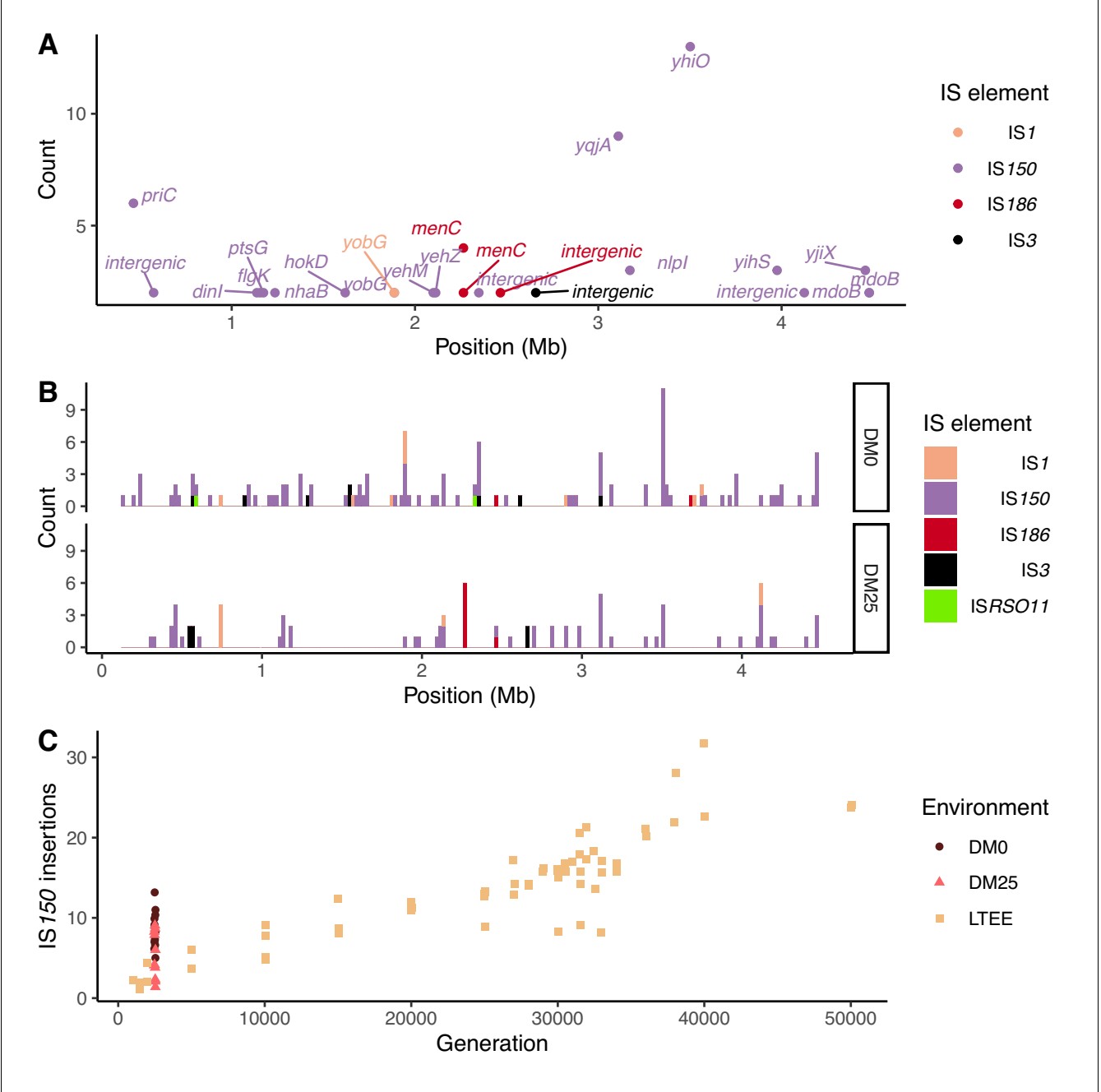

**Figure 12.** Parallel IS-element insertions. (**A**) Counts of parallel IS-element insertions in labeled genes (including promoter and coding regions) summed across sequenced DM0- and DM25-evolved genomes, and arranged by position on the *E. coli* chromosome, relative to the inferred last common ancestor of all strains (Materials and methods). IS1 insertions are shown in pink, IS150 in lavender, IS186 in red, IS3 in black, and ISRSO11 in green. Some genes contain multiple sites with parallel IS-element insertions. (**B**) Location of insertions, shown separately for the DM0- and DM25-evolved genomes. Colors are the same as in panel A. (**C**) Total number of IS150 insertions in the DM0- and DM25-evolved genomes after 2500 generations. The corresponding numbers of IS-element insertions in clones isolated from LTEE population Ara−3 at time points over 50,000 generations of evolution are shown for comparison. DM0 clones are labeled as brown circles, DM25 clones as pink triangles, and LTEE Ara−3 clones as tan squares.

The online version of this article includes the following source data for figure 12:

**Source data 1.** Table of IS-element insertions in evolved genomes.

**Source data 2.** Table of IS-element insertions in LTEE and Mutation Accumulation Experiment (MAE) genomes, originally published in *Tenaillon et al., 2016*.

Insertions of IS*150* into new sites were strongly parallel across the independently evolved populations within, but not between, the two environments (*Figure 12A and B*). These systematic differences led us to hypothesize that the parallel IS insertions reflect the influence of selection, rather than insertion-site biases (*Figure 11*, *Figure 12B*; *Tenaillon et al., 2016*). We evaluated this hypothesis by conducting a randomization test for selection-driven parallel IS*150* insertions over and above a null model that assumes only insertion-site preferences (Materials and methods). The most extreme observed case of parallelism at the base-pair level was an IS*150* insertion at the −35 position of the promoter for *yhiO*, which encodes the universal stress protein UspB, which happened in 9 of the 12 DM0 genomes (randomization test with 100,000 bootstraps: p=0.014). This test is even more conservative because it excludes two other IS*150* insertions affecting this same gene in the DM0 genomes: an IS*150* insertion at the −36 position of the promoter and another IS*150* insertion in *yhiO* itself. Therefore, we can reject the null hypothesis that site-specific insertion biases alone provide an adequate explanation for the distribution of IS*150* insertions. Given the conservative nature of this test, it is quite possible that some other parallel IS-insertions also indicate positive selection.

## Parallel amplification mutations in the DM0- and DM25-evolved populations

We detected tandem amplifications of large genomic regions, often to high copy-number, in many DM0- and DM25-evolved clones (*Tables 1* and *2*, *Figure 13*). All genomes include amplifications containing the novel genetic module that evolved during the LTEE, which places one or more copies of the *citT* gene under the control of the *rnk* promoter region, with the exception of the anomalous Cit⁻ clone ZDBp874 (*Table 1*). This new *rnk-citT* module provides access to citrate, and mutations that increase its dosage improve growth on citrate (*Blount et al., 2012*; *Van Hofwegen et al., 2016*).

Other amplifications include the *dctA* gene (*Table 2*). DctA is a proton motive force-driven, generalized di- and tricarboxylic acid transporter. During growth on citrate, the CitT antiporter protein exports TCA cycle intermediates into the medium in exchange for citrate. DctA enables recovery of those intermediates. Two mechanisms of increasing *dctA* expression have been shown to improve growth on citrate. *Quandt et al., 2014* identified mutations in the *dctA* promoter that cause high-level expression. *Van Hofwegen et al., 2016* showed that increased copy number of *dctA* is likewise beneficial. We found evidence that these two mechanisms are anticorrelated. Two of the ancestral clones, CZB151 and CZB154, have a shared (identical by descent) mutation in the promoter sequence of *dctA*. The third ancestor, CZB152, lacks this mutation. Only one of the 16 evolved descendants of CZB151 or CZB154 has a *dctA* amplification, whereas five of CZB152's eight descendants have such an amplification (Fisher's exact test: p=0.0069). Also supporting this anticorrelation, one of the three CZB152 descendants without a *dctA* amplification independently evolved a mutation affecting that gene's promoter.

We identified another set of parallel amplifications in six evolved genomes. These amplifications are large and highly variable in extent, but all include at least the *fdnI*, *yddM*, *adhP*, *maeA*, *rpsV*, and *bdm* genes. These amplifications were often present in high copy numbers. Three DM25-evolved genomes have 2–13 copies, and three that evolved in DM0 have 28–59 copies (*Table 2*). In one case, ZDBp889, the amount of DNA in the amplified region constitutes more than 15% of the total evolved genome (*Figure 13*). By contrast, the amplifications of *citT* and *dctA* contain an average of 4–5 and 2–3 copies, respectively (*Tables 1* and *2*).

These long, high-copy-number amplifications must exert a metabolic burden, due to the costs of additional DNA synthesis and increased gene expression (*da Silva and Bailey, 1986*; *Lenski and Nguyen, 1988*). The repeated evolution of amplifications of this genomic region suggests that they confer some selective benefit that outweighs their cost. We examined the genes shared among the amplifications to identify which might confer this benefit. The *rpsV* gene, which encodes the 30S ribosomal subunit protein D, appears to have been a minor target for adaptation to DM0 based on parallel mutations (*Figure 11*). The *maeA* gene encodes an NAD⁺-dependent oxaloacetate-decarboxylating malate dehydrogenase (EC 1.1.1.38) that catalyzes the decarboxylation of malate to pyruvate. This plausible connection to citrate metabolism led us to hypothesize that increased *maeA* dosage and expression provides the benefit that overcomes the cost imposed by the amplifications.

**Table 1.** Copy number of amplified *citT* genes in sequenced clones.

| Genome | Medium | Mean copy number | Minimum copy number[*] | Maximum copy number[*] |
|---|---|---|---|---|
| CZB151 | DM25 | 4.21 | 3.39 | 5.47 |
| CZB152 | DM25 | 8.23 | 5.17 | 11.46 |
| CZB154 | DM25 | 4.14 | 1.72 | 9.83 |
| ZDBp871 | DM0 | 2.82 | 1.70 | 4.26 |
| ZDBp875 | DM0 | 11.37 | 8.05 | 14.93 |
| ZDBp877 | DM0 | 7.68 | 3.66 | 11.33 |
| ZDBp880 | DM0 | 3.82 | 1.79 | 5.93 |
| ZDBp883 | DM0 | 5.08 | 1.88 | 12.81 |
| ZDBp886 | DM0 | 4.76 | 2.27 | 6.88 |
| ZDBp889 | DM0 | 4.66 | 2.90 | 7.11 |
| ZDBp892 | DM0 | 5.69 | 2.21 | 8.76 |
| ZDBp895 | DM0 | 5.30 | 2.13 | 8.51 |
| ZDBp898 | DM0 | 6.14 | 2.77 | 9.73 |
| ZDBp901 | DM0 | 3.91 | 1.83 | 5.64 |
| ZDBp904 | DM0 | 3.84 | 1.78 | 5.68 |
| ZDBp910 | DM25 | 4.71 | 2.40 | 6.47 |
| ZDBp911 | DM25 | 3.13 | 1.58 | 5.01 |
| ZDBp912 | DM25 | 8.93 | 4.51 | 13.41 |
| ZDBp913 | DM25 | 4.83 | 2.68 | 6.94 |
| ZDBp914 | DM25 | 4.11 | 2.03 | 6.17 |
| ZDBp915 | DM25 | 3.31 | 1.78 | 5.19 |
| ZDBp916 | DM25 | 4.87 | 2.67 | 6.95 |
| ZDBp917 | DM25 | 3.20 | 1.77 | 4.63 |
| ZDBp918 | DM25 | 3.66 | 1.88 | 5.18 |
| ZDBp919 | DM25 | 2.91 | 2.06 | 3.81 |
| ZDBp920 | DM25 | 3.92 | 2.08 | 5.49 |
| ZDBp921 | DM25 | 5.76 | 2.93 | 9.15 |

[*]These bounds indicate the ratio of the minimum and maximum sequencing coverage measured at the *citT* locus to the mean coverage over the genome. In all cases, the estimated copy number is significantly greater than one at $p < 0.0001$, even after Bonferroni corrections for multiple tests of the same hypothesis.

The online version of this article includes the following source data for Table 1:

**Source data 1.** Copy number of amplifications affecting *citT*, *dctA*, and *maeA* in the ancestral and evolved clones.

## Increased MaeA expression is highly beneficial in the citrate-only environment

We tested our hypothesis that increased *maeA* dosage confers a fitness benefit by transforming the ancestral strains CZB151 and CZB152 with a low-copy plasmid, RM4.6.2, which contains a copy of *maeA* that is under the control of a strong constitutive synthetic promoter and ribosome-binding site. These Ara⁻ RM4.6.2 transformants were competed in DM0 against Ara⁺ mutants (ZDB67 and ZDB68, respectively) of the same clones transformed with the empty-plasmid control. The RM4.6.2 transformants had a fitness advantage of ~28% in both the CZB151 (*n* = 6; mean fitness = 1.2790, *t*-distributed 95% confidence interval: [1.2636, 1.2944]) and CZB152 (*n* = 6; mean fitness = 1.2778, *t*-distributed 95% confidence interval: [1.2597, 1.2959]) backgrounds relative to their otherwise iso-genic competitors. Overexpression of *maeA* is therefore highly beneficial in the DM0 environment, and its benefit likely explains the high-copy-number amplifications containing *maeA* found in many evolved clones.

**Table 2.** Copy number of amplified *maeA* and *dctA* genes in sequenced clones from populations that evolved for 2500 generations in either DM0 or DM25 environments.

| Genome | Medium | Gene | Mean copy number | Minimum number[*] | Maximum number[*] | Adjusted *p*-value[†] |
|---|---|---|---|---|---|---|
| ZDBp880 | DM0 | *dctA* | 2.33 | 1.67 | 3.26 | <0.0001 |
| ZDBp886 | DM0 | *dctA* | 3.20 | 1.88 | 4.25 | <0.0001 |
| ZDBp898 | DM0 | *dctA*[‡] | 2.09 | 1.71 | 2.76 | 0.0023 |
| ZDBp913 | DM25 | *dctA* | 3.17 | 1.91 | 4.68 | <0.0001 |
| ZDBp918 | DM25 | *dctA* | 3.41 | 1.80 | 5.19 | <0.0001 |
| ZDBp919 | DM25 | *dctA* | 2.60 | 2.06 | 3.29 | <0.0001 |
| ZDBp883 | DM0 | *maeA* | 58.47 | 22.61 | 95.46 | <0.0001 |
| ZDBp889 | DM0 | *maeA* | 34.71 | 2.35 | 55.39 | <0.0001 |
| ZDBp904 | DM0 | *maeA* | 28.08 | 15.30 | 44.72 | <0.0001 |
| ZDBp911 | DM25 | *maeA* | 2.22 | 1.54 | 3.39 | <0.0001 |
| ZDBp917 | DM25 | *maeA* | 4.72 | 3.56 | 6.18 | <0.0001 |
| ZDBp919 | DM25 | *maeA* | 12.81 | 2.16 | 18.00 | <0.0001 |

[*]These bounds indicate the ratio of the minimum and maximum sequencing coverage measured at the indicated locus to the mean sequencing coverage over the genome.

[†]Significance levels are shown after Bonferroni corrections for multiple tests of the same hypothesis.

[‡]There may be two discontinuous amplifications of *dctA* in this genome, or there may be a single continuous amplification with a short region of low coverage within the gene. The second region of amplification has a similar copy number. We present data for only one region, which provides a conservative estimate of the overall statistical significance in this case.

The online version of this article includes the following source data for Table 2:

Source data 1. Copy number of amplifications affecting *citT, dctA, and maeA* in the ancestral and evolved clones.

We used RNA-Seq to verify that clones with *maeA*-containing amplifications have elevated transcription of that gene. We compared the transcriptomes of two ancestral clones (CZB151 and CZB152) and two DM0-evolved clones with *maeA* amplifications (ZDBp883, ZDBp889). Both evolved clones do, indeed, have much higher levels of *maeA* expression than their respective ancestors (*Figure 14*).

Despite the large fitness advantage conferred by increased *maeA* dosage in the citrate-only DM0 environment, most of the evolved Cit⁺ genomes we examined do not have *maeA* amplifications. Moreover, although three of the DM25-evolved genomes in this study have large *maeA* amplifications (*Table 2*), none have been found in the sequenced genomes of Cit⁺ clones isolated from the Ara−3 parent population in the LTEE itself. This discrepancy might be explained by the evolution of increased *maeA* expression via other mutations. To evaluate this possibility, we also used RNA-Seq to examine the transcriptome of ZDBp877, a DM0-evolved clone without a *maeA* amplification. In contrast to ZDBp883 and ZDBp889, ZDBp877 expresses *maeA* at a level similar to that of the ancestral clones (*Figure 14*). This finding means that at least some, and perhaps all, of the evolved clones without *maeA* amplifications lack mutations that boost its expression.

## Transcriptomic analysis of DM0-evolved clones

We identified other potentially adaptive differences in transcription between the DM0-evolved clones during growth in the DM0 medium (*Figure 14*). The two evolved clones with *maeA* amplifications, ZDBp883 and ZDBp889, both show increased expression of the *fad* fatty acid β-oxidation regulon, whereas ZDBp877, which lacks a *maeA* amplification, does not. ZDBp877 and ZDBp889, but not ZDBp883, both downregulate the cytochrome $bo_3$ terminal oxidase complex, *cyoABCD*. We also found three genes with more extreme differential expression than *maeA* between the clones with and without the *maeA* amplification. These genes are *dinI, gltS,* and *ECB_03510*, all three of which are strongly downregulated in ZDBp877 in comparison to both clones with the amplification. DinI is a DNA-damage inducible protein that regulates the SOS response. GltS is a glutamate/sodium

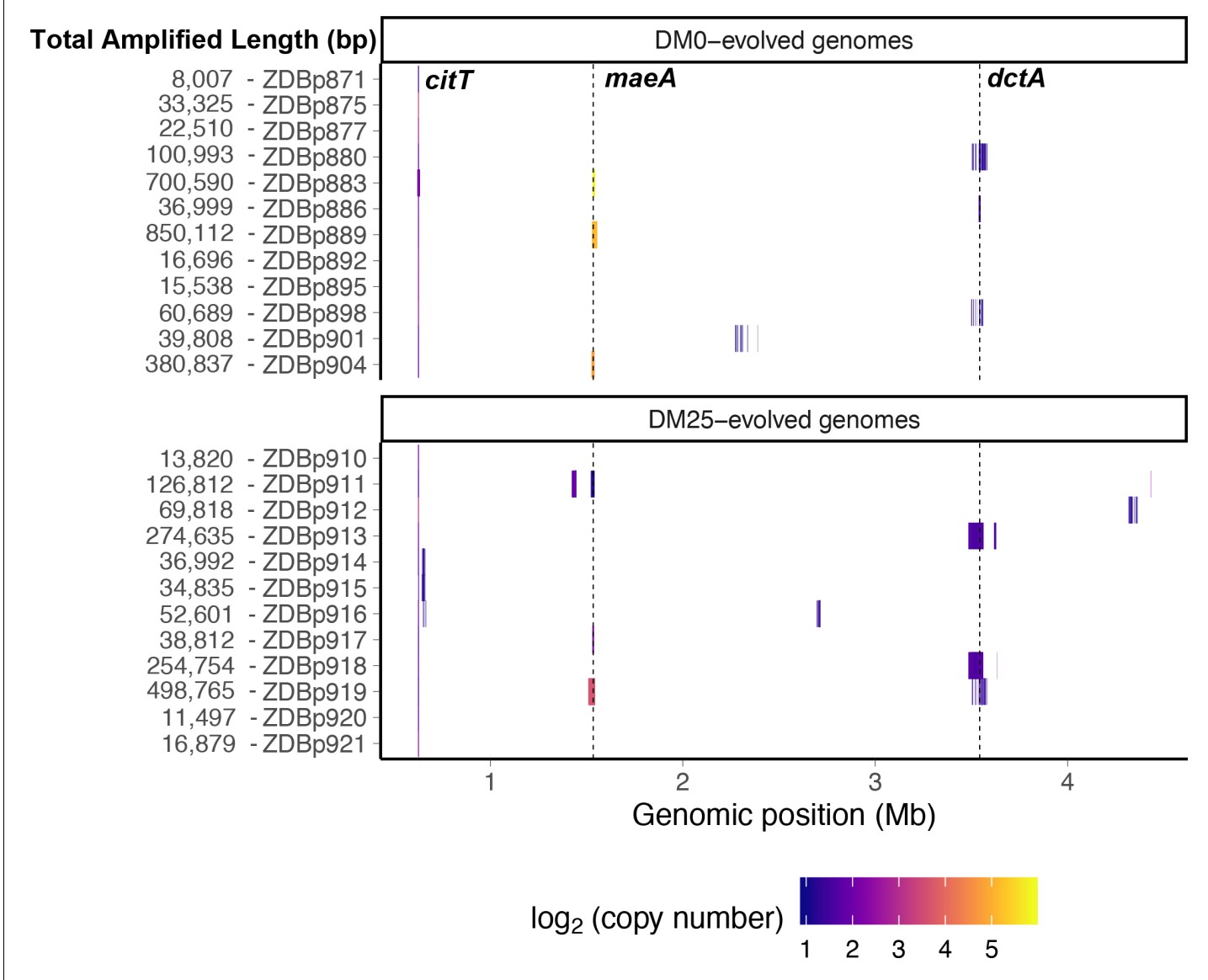

**Figure 13.** Genetic amplifications in evolved clones. Genomic regions with significant amplifications in DM0- and DM25-evolved clones, arranged by chromosomal position. The evolved clones from DM0 (top half) and DM25 (bottom half) are indicated at the near left, with the total amplified length shown at the far left. Dashed vertical lines mark the *maeA* and *dctA* loci. The boundaries vary among the subset of genomes with amplifications that encompass these genes; by contrast, the *citT* locus is amplified in all of these genomes, and with nearly uniform boundaries. Colors denote amplification copy-number on a log$_2$ scale from dark (low copy-number) to light (high copy-number).

The online version of this article includes the following source data for figure 13:

**Source data 1.** Table of amplifications discovered by examining sequencing coverage in evolved genomes.

symporter, while *ECB_03510* encodes a protein of unknown function that lies immediately downstream of *gltS*.

These differences aside, we found largely similar changes in gene expression across the three DM0-evolved clones relative to their ancestors (*Figure 14*). All three display strong downregulation of the UspB stress protein encoded by *yhiO*, presumably caused by the parallel IS*150* insertions into that gene's promoter. We also found extensive downregulation of genes encoding ribosomal proteins (including *rpsB*, *rpsU*, *rpsO*, *rpsT*, *rplE*, *rplJ*, *rplN*, and *rplX*); genes involved in RNA transcription (*rpoA*, *rpoB*, *rpoC*, *rpoS*, *rho*); and DNA-replication associated genes (*gyrA*). Other downregulated genes in the evolved clones include the *nuo* operon, which encodes NADH

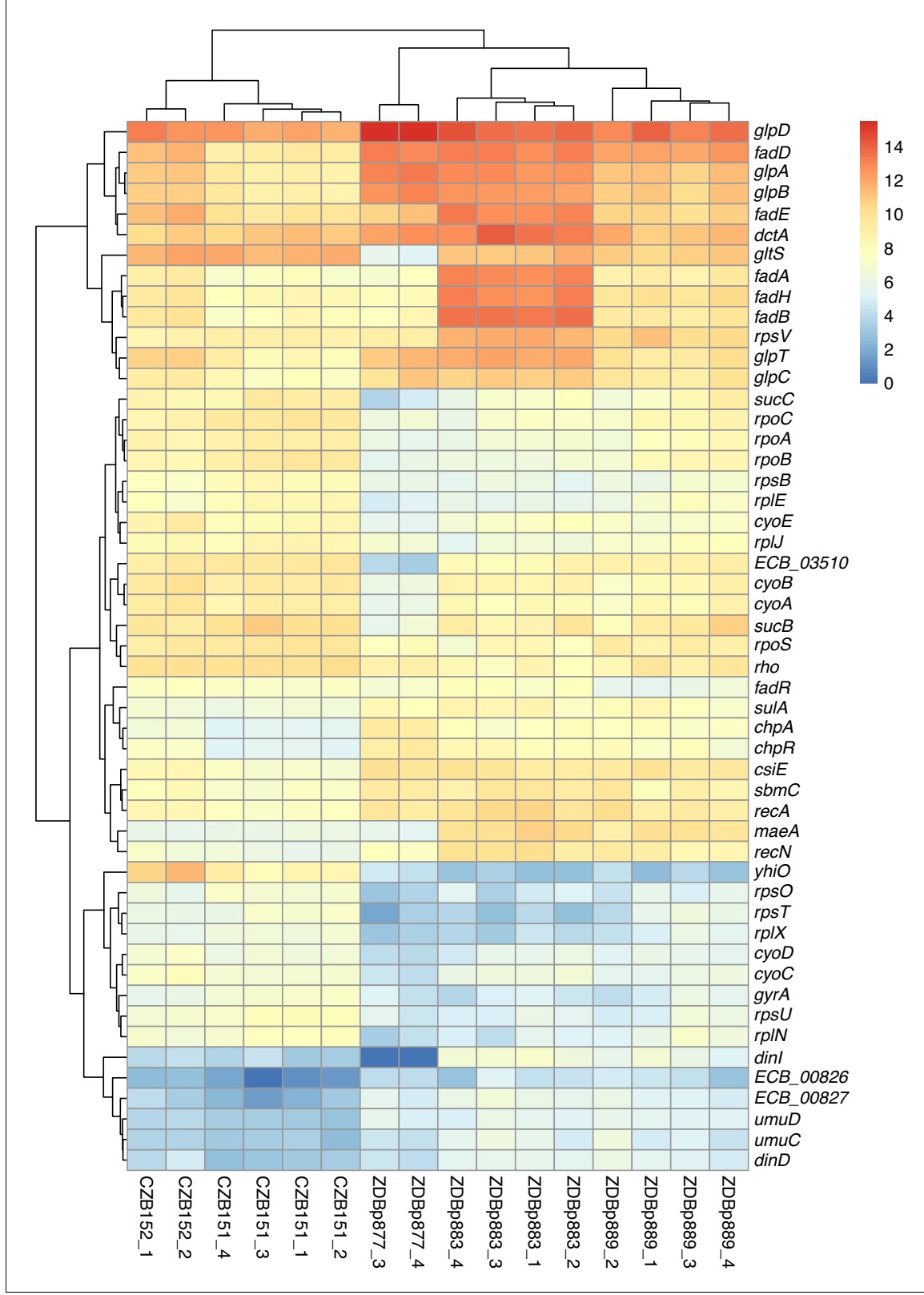

**Figure 14.** Transcriptomic analysis of ancestral and evolved clones. Differential expression analysis comparing two ancestral (CZB151 and CZB152) and three evolved clones (ZDBp877, ZDBp883, ZDBp889), produced by *sleuth* (*Pimentel et al., 2017*). The colored bar (at right) shows the level of RNA expression based on estimated counts and transformed as log$_2$(1 + est_counts). The differentially expressed genes discussed in the main text are

*Figure 14 continued on next page*

*Figure 14 continued*

shown here. The numeric labels after the strain identifiers indicate the two or four biological replicates for each clone (i.e., RNA samples prepared from independently revived cultures of that clone).

The online version of this article includes the following source data for figure 14:

**Source data 1.** RNA transcript quantification using *kallisto* software.

dehydrogenase in the respiratory electron transport chain, and key TCA cycle genes including those encoding the 2-oxoglutarate dehydrogenase complex (*sucB*, *sucC*, and *lpdA*). By contrast, we see strong upregulation of genes encoding certain prophage-associated proteins (ECB_00826 and ECB_00827); some toxin-antitoxin pairs (*chpA-chpR*); proteins involved in recombinational DNA repair (*recA* and *recN*); SOS response proteins (*dinD*, *sulA*, *umuC*, and *umuD*); proteins associated with stationary phase (*csiE* and *sbmC*); a biofilm-associated stress protein (*bhsA*); and others of unknown significance (*Supplementary file 3*).

The downregulation of transcription, translation, and NADH dehydrogenase genes and the increased expression of stress-associated genes suggest that adaptation to DM0 involved reducing growth rate (relative to the faster growth on glucose), presumably to achieve balanced growth on citrate alone. The upregulation of fatty-acid β-oxidation genes (albeit less so in the ZDBp877 clone without the *maeA* amplification) also indicates some remodeling of the connection between fatty acid and citrate metabolism that is mediated by acetyl-CoA. Other changes, including the upregulation of the *fad* operon encoding fatty acid degradation in ZDBp883 and ZDBp889, the anaerobic glycerol-3-phosphate dehydrogenase operon (*glpABCD*) in ZDBp877 and ZDBp889, and the glycerol-3-phosphate transporter (*glpD* and *glpT*) in ZDBp883 (*Supplementary file 3*), suggest adaptation to scavenging on dead and dying cells in the DM0 populations.

## Discussion

It is rarely feasible to examine evolution in action as organisms invade, colonize, and adapt to a new niche in nature, especially with independently evolving replicates and control populations. In this study, we investigated how *E. coli* variants with the new ability to grow aerobically on citrate adapted to a novel, citrate-only resource environment in the laboratory. We examined the genomic and phenotypic evolution of 12 initially clonal populations after 2500 generations in this new environment, along with 12 initially identical control populations maintained for the same time in the ancestral environment that contains glucose as well as citrate, to better understand their post-invasion potential, including refinement of the Cit⁺ trait.

The founding clones grew poorly in their new medium, exhibiting long lag phases, slow growth, and high variation in their growth kinetics. However, those founding clones had substantial potential to adapt to citrate as their sole carbon and energy source, as the experimental populations evolved shorter lag times and faster growth rates in DM0. The evolved populations also showed correlated improvements in the ancestral glucose-citrate medium, DM25. These changes are consistent with selection pressures typical for evolution experiments that use a serial batch-culture regime like that of the LTEE (*Vasi et al., 1994*), upon which our experiment was based. In contrast to the LTEE, but consistent with other lines of evidence that growth on citrate is stressful for *E. coli*, the growth-curve trajectories and even stationary-phase optical densities exhibited substantial variability for replicate assays performed using the same population sample. Assays of competitive fitness over the same 24 hr transfer cycle also showed extreme variability, especially in DM0, making it difficult to reliably estimate overall fitness gains. Increasing the duration and replication of the fitness assays should help reduce this variation in future work. Nonetheless, our difficulty in measuring adaptation in this system is striking in contrast to the ease of doing so in the LTEE (*Lenski et al., 1991*; *Wiser et al., 2013*). It is also possible that other demographic components of fitness besides shorter lag times and faster growth rates are at play in this citrate-based system. Indeed, we observed extensive cell death in Cit⁺ clones, and the level of mortality varied considerably even between replicate assays for reasons that we do not understand.

The evolved clones exhibit even greater variation in their growth phenotypes. While most have faster growth rates and shorter lag times, similar to the improvements observed at the population level, some evolved clones grow only slightly better, or even worse, than their ancestors. Similarly,

most evolved clones show no significant increase in fitness, and some are less fit than their ancestor, even in the environment where they evolved. In some cases, the fitness estimates are discordant with measured growth characteristics. For example, one clone from a population that evolved in DM25 (ZDBp917) has a lag time far longer than its ancestor, yet it has a marginally higher fitness.

Complex ecological interactions between genotypes might explain such discordant outcomes between competitive fitness assays and growth parameters estimated in pure culture. For example, non-transitive competitive interactions can give rise to evolved clones that are more fit than their immediate predecessors, but less fit than their earlier ancestors (*Paquin and Adams, 1983*; *Buskirk et al., 2019*). We cannot exclude the possibility of non-transitive dynamics in our system at this time, but we note that they have not been observed in the LTEE on which our experiments are based (*de Visser and Lenski, 2002*; *Lenski, 2017*). A more likely alternative is that the paradoxical changes in fitness and growth are caused, in part, by cross-feeding and similar negative frequency-dependent interactions. The ancestral clones, for example, might be better at invading some of the evolved communities than they are at growing alone in the same medium. Similarly, some evolved clones with paradoxical growth phenotypes might be specialized ecotypes that have adapted to unknown niches, such as scavenging on dead cells or cross-feeding on metabolites produced by coexisting lineages (*Turner et al., 1996*; *Rozen et al., 2009*; *Velicer and Mendes-Soares, 2009*; *Le Gac et al., 2012*; *Maddamsetti et al., 2015*; *Good et al., 2017*). We will investigate the possibility of complex ecological interactions in future work.

Genomic plasticity reflecting copy-number variation and transposable-element activity played a key role in adaptation to the citrate niche. These findings support and extend previous work showing the importance of such plasticity in adaptation to other selective challenges (*Chang et al., 2013*; *Vandecraen et al., 2017*; *Press et al., 2019*; *Lauer and Gresham, 2019*), including the rapid evolution of antibiotic heteroresistance in some pathogens (*Nicoloff et al., 2019*). Our work especially bolsters previous demonstrations of the evolutionary importance of dynamic gene amplifications, which increase the dosage of genes encoding specific products needed at higher levels (*Patrick et al., 2007*; *Andersson and Hughes, 2009*). For example, *Blank et al., 2014* found that *E. coli* strains with single-gene knockouts rapidly re-evolved the capacity to grow in minimal medium in part via amplifications that increased genome size by more than 20%. Such amplifications may also impose substantial metabolic costs, and they are prone to recombination-mediated collapse, so they are readily lost when the relevant gene products are no longer needed at higher levels. Amplifications also increase the opportunity for further mutations that may provide a benefit for a single copy, thereby favoring subsequent collapse and elimination of the cost of multiple copies (*Andersson and Hughes, 2009*; *Brennan et al., 2015*; *Näsvall et al., 2012*). In addition to their role as substrates for promoting amplifications, IS elements seem to have both inactivated and modulated the expression of various genes in our evolution experiment. The activity of some IS elements appears to be increased by stress, which cells may experience when they invade a new niche to which they are poorly adapted (*Vandecraen et al., 2017*). Of course, this plasticity is a double-edged sword: the genomic instability that transposable elements cause can also produce deleterious mutations, which could impede adaptation and might even lead to extinction, especially when small founding populations invade a new niche.

Altogether, our results have several nuanced implications for evolution following the innovation-driven discovery of new niches. They suggest that genomes often possess latent potential to refine novel traits and adapt to new niches. This potential can be fulfilled not only by point mutations, but also via larger mutations such as gene amplifications and transpositions that may allow more rapid adaptation after niche discovery. Despite such adaptation, however, suboptimal traits may persist long after the new niche has been successfully invaded. In our study, the evolved Cit$^+$ bacteria's physiology shows an evolutionary mismatch with their growth on the newly accessible citrate, even after thousands of generations of adaptation to that new niche. Evidence for the mismatch includes erratic growth trajectories and fitness measures that suggest extreme sensitivity to small differences in the environment, the identity of competitors, or both. It is also seen in the high levels of mortality during stationary phase in some Cit$^+$ clones, as shown using live-dead staining of cells. Further evidence comes from analyses of transcriptomic data, which shows that some Cit$^+$ lines evolved increased expression of stress-associated genes during exponential phase. Thus, while a nascent ecotype's latent potential for adaptation may allow its establishment in a new niche (especially in the

absence of established competitors), it may nonetheless continue to experience stress and suffer from suboptimal phenotypes for a long time before becoming truly well suited to its new conditions.

Our findings also highlight that the fact that organisms are historically contingent patchworks of traits and functions constructed by evolutionary tinkering, and so they are rife with design compromises (*Pittendrigh, 1958*; *Tinbergen, 1965*; *Jacob, 1977*). Natural selection integrates these complex assemblies into idiosyncratic, but usually robust and stable, biological systems. This stability can be disrupted, however, such as when a novel trait that is beneficial, on the whole, nonetheless generates new stresses. These secondary tradeoffs are also implicit in Fisher's geometric model of adaptation, in which mutations of large effect, including those that produce new functions, are especially likely to disrupt other phenotypes (*Fisher, 1930*; *Orr, 2000*). We have shown that such disruptions can persist for thousands of generations, which implies that re-evolving a stable, robust system in which a novel trait is fully integrated with the preexisting physiology can be difficult.

Organisms typically maintain their physiological systems at a dynamic steady state. This homeostasis implies that organisms have evolved to maintain physiological variables within an acceptable range in the face of perturbations (*Albergante et al., 2014*). Failure to maintain homeostasis may result in illness and even death. Viability is sometimes possible outside the usual range of homeostasis, but often at the cost of stress and lasting damage to the organism. Our findings imply that the disruptions caused by evolutionary novelties can maladaptively change the system parameters that maintain homeostasis, thereby causing stress and increasing mortality. Many questions remain about the nature and consequences of these homeostatic disruptions, as well as how novel traits might eventually become well integrated with an organism's existing physiology to restore its prior homeostasis.

Our experimental system has the potential to address these and other questions about innovation, adaptation, and maladaptation that are relevant to both evolutionary biology, in general, and evolutionary medicine, in particular. In humans, for example, cultural innovations, including the agricultural revolution, have vastly reshaped our diets and thereby also changed our gut microbiota (*McMichael et al., 2007*; *David et al., 2014*). Contemporary high-calorie diets and sedentary lifestyles have led to an epidemic of associated illnesses, including hypertension, diabetes, and obesity. Might new traits typically exhibit more phenotypic variation, reflecting greater sensitivity to intrinsic stochasticity, lower robustness to environmental perturbations, or both? If growth on citrate by *E. coli* required major changes in physiology and metabolism, then that innovation may have increased fragility due to new difficulties in coordinating cell growth and division (*Scott et al., 2014*; *Schaechter, 2015*). By disrupting existing physiological and metabolic processes, innovations can introduce new compromises and imbalances, the resolution of which requires novel variation. That new variation may, in turn, affect correlated traits and the organism's overall robustness. We conjecture that the evolutionary refinement of traits that open new niches may often promote evolvability at the expense of robustness and overall good health (*Lenski et al., 2006*).

## Materials and methods

Key resources are listed and described in *Supplementary file 1*: Key Resources Table.

### Evolution experiment

We previously isolated three random Cit$^+$ clones, designated as CZB151, CZB152, and CZB154, from the 33,000-generation sample of LTEE population Ara−3 (*Blount et al., 2008*). We also isolated spontaneous Ara$^+$ revertants for each clone, designated as ZDB67, ZDB68, and ZDB69, respectively. For long-term preservation, we inoculated Luria Bertani (LB) broth with isolated colonies of each clone and its revertant, grew them overnight at 37°C with orbital shaking at 120 rpm, and froze samples of each at −80°C with glycerol as cryoprotectant. We revived the clones and revertants from the frozen stocks and grew them in LB overnight. We then diluted the LB cultures 10,000-fold into 9.9 mL of Davis Mingioli (DM) minimal medium supplemented with 25 mg/L glucose (DM25), and grew them at 37°C with orbital shaking at 120 rpm. After 24 hr, we diluted these cultures 100-fold in 9.9 mL of fresh DM25 and grew them for another 24 hr. This preconditioning acclimated the bacteria to growing on citrate. The preconditioned cultures were then diluted 100-fold into 9.9 mL of base DM medium (DM0), which lacks any glucose but contains 1 g/L (1,700 mM) of citrate for carbon and energy. We started two replicate populations from each LTEE-derived clone and each

revertant, for a total of 12 DM0 populations. At the same time, we inoculated 12 populations into DM25 (*Figure 1*). We maintained these DM25 populations at 37°C with orbital shaking, and transferred them by 100-fold dilution into fresh DM25 every 24 hr (i.e., the same conditions as in the LTEE) for 375 transfers and 2500 generations in total. The founding Cit+ clones grow poorly in the citrate-only resource environment. They were unable to reach stationary phase or, in some cases, exponential phase within 24 or even 48 hr. We therefore incubated the DM0 populations for 72 hr after their initial inoculation so they could reach stationary phase before transfer to fresh medium. We then diluted them 100-fold into 9.9 mL of DM0 every 48 hr for seven cycles (two weeks), and then subsequently every 24 hr for a total of 375 transfers and 2500 generations. Every 37 days (~250 generations) samples of each population were frozen with glycerol at −80°C.

## Isolation of evolved clones

We revived each evolved population sample by inoculating 100 µL of the stock frozen at generation 2500 into 9.9 mL of LB broth and incubating overnight at 37°C with orbital shaking. We then diluted the revived DM0- and DM25-evolved populations 10,000-fold in 9.9 mL of DM0 or DM25, respectively, and grew them for 24 hr at 37°C with orbital shaking, followed by 100-fold dilution into fresh DM0 or DM25 and another 24 hr period of growth at 37°C with orbital shaking. We then diluted each population 100,000-fold in 0.85% saline and spread 100 µL on an LB agar plate marked with three dots on the bottom. We streaked the colony closest to each dot on an LB plate after 48 hr of incubation at 37°C, thereby providing three randomly chosen clones from each population. We then inoculated an isolated colony of each clone into LB broth, grew it overnight, and froze it as before.

## Fitness assays

We measured fitness by performing competition experiments modified from those described by *Lenski et al., 1991*. We revived samples by inoculating 15 µL (for clones) or 100 µL (for whole populations) from a slightly thawed frozen stock into 10 mL of LB. These cultures then grew overnight at 37°C with 120 rpm orbital shaking, after which we diluted each 10,000-fold into either DM25 or DM0 and preconditioned as described above. We inoculated 50 µL of each competitor's preconditioned culture into 9.9 mL of the corresponding medium, vortexed to mix, and then we spread 100 µL of $10^{-2}$ and $10^{-3}$ dilutions on Tetrazolium Arabinose (TA) indicator agar plates to estimate the competitors' initial densities. We estimated their densities again at the end of the assay by spreading 100 µL of $10^{-4}$ and $10^{-5}$ dilutions on TA plates. For whole populations, we assayed fitness with 3-fold replication in one-day competitions, in which final densities were estimated after 24 hr. For the evolved clones, we assayed fitness with 5-fold replication, and measured final densities after 3 days, with 100-fold serial transfers to fresh medium after 24 and 48 hr. The realized growth rates of the two competitors were determined from their starting and ending densities, accounting for the dilutions. We calculated the fitness of an evolved clone or population as its realized growth rate divided by that of the ancestral competitor. In the population fitness assays, ZDB67 was the common competitor for all Ara− population samples, and CZB151 was the common competitor for all Ara+ population samples.

## Growth curves

We chose one of the three evolved clones from generation 2500 from each DM0 or DM25 population, then revived and preconditioned it in DM0 or DM25 as described above. We diluted the cultures 100-fold into 9.9 mL of DM0 or DM25, vortexed, and dispensed six 200 µL aliquots of each culture into wells in a 96-well plate. We randomized well assignments for the cultures to minimize position effects. We measured optical density (OD) at 420 nm wavelength every 10 min for 48 hr using a Molecular Devices SpectraMax 384 automated plate reader. We discarded the measurements taken before 30 min from our analysis.

## Microscopy and cell viability analyses

We performed microscopy and viability analyses on cells derived from five clones: the LTEE ancestor (REL606); one of the three Cit+ ancestors in our evolution experiment (CZB151); two of its descendants that evolved in DM0 and DM25 for 2500 generations (ZDBp871 and ZDBp910, respectively); and a Cit+ clone isolated at generation 50,000 of the LTEE (REL11364). We revived clones from the

frozen stocks and preconditioned them as described above, except that the preconditioning steps in DM0 or DM25 were extended to four daily passages to ensure acclimation to these environments. We performed preparations for live/dead cell staining and microscopic analyses on the fifth day. In these preparations, we concentrated the cells in each culture by centrifugation at 7,745 g for 8 min and decanted the supernatant. We then resuspended the cell pellets in Corning tubes containing 10 mL of 0.85% saline, and incubated them at room temperature for 1 hr; we inverted the tubes every 15 min. We then centrifuged these cultures for an additional 8 min, decanted the supernatant, and resuspended the cell pellets in 0.85% saline. We adjusted the volume of saline based on variation in turbidity to ensure that we had sufficient cells in a typical field of view for microscopy. We examined 14–55 fields per replicate for each combination of strain and media treatment. Total cell counts ranged from approximately 15,000 to 60,000 for the various combinations of clones and culture media.

We used the LIVE/DEAD *Bac*Light Viability Kit for microscopy (ThermoFisher #L7007), following the manufacturer's directions for fluorescently labeling cells. In short, we mixed components A and B in equal amounts, added 1 µl to each culture containing resuspended cells, and incubated them for 20 min in the dark to prevent photobleaching. After labeling, we fixed 3 µL of each sample onto a 1% agarose pad and performed fluorescent microscopy using a Nikon Eclipse Ti inverted microscope. Phase-contrast images were taken using diascopic illumination with an exposure time of 100 ms. Fluorescence was measured with an exposure time of 200 ms at 25% power of the fluorescent light source using two filter sets, 49003-ET-EYFP and 49008-ET-mCherry Texas Red (Chroma), which correspond to the fluorescence spectra of 'live' and 'dead' cells, respectively. All images were taken at $100\times$ magnification.

We analyzed micrographs using *SuperSegger*, an image-processing package (*Stylianidou et al., 2016*). We first filtered the data, keeping only those values for segmented regions in the micrograph that were scored by the neural-network classifier as having $P$(Cell = True) > 75%. (Region scores range between $-50$ and 50, so we used data only from regions with values between 25 and 50). We then used the fluorescence values from the *SuperSegger* output and scored individual cells as 'live' or 'dead' depending on whether the fluorescence signal on the green (YFP) channel was greater or lesser, respectively, than the signal on the red (RFP) channel. We calculated the proportion of dead cells across the many fields examined for each of the five replicate cultures that we analyzed for each combination of clone and growth medium, and we used these values in the statistical analyses.

## Genomic analysis and copy-number variation

We thawed the 3 Cit⁺ founder strains (CZB151, CZB152, CZB154), their respective Ara⁻ derivatives (ZDB67, ZDB68, ZDB69), and 25 evolved clones (one Cit⁺ clone from each DM0 and DM25 evolved population, plus the anomalous Cit⁻ clone ZDBp874) and grew them overnight in LB broth. We isolated genomic DNA from each sample using the Qiagen Genomic-tip 100/G DNA extraction kit. The genomic DNA was then sequenced using the platforms in the core facilities shown in *Supplementary file 4*.

For genomes sequenced at UT Austin, we purified DNA from *E. coli* cultures using the PureLink Genomic DNA Mini Kit (Invitrogen). For each sample, we fragmented 1 µg of purified DNA using dsDNA Fragmentase (New England Biolabs). We then used the KAPA Low Throughput Library Preparation kit (Roche) to construct Illumina sequencing libraries according to the manufacturer's instructions with two exceptions. First, we reduced reaction volumes by half. Second, we designed DNA adapters that incorporate additional 6-base sample-specific barcodes such that the barcodes are sequenced as the first bases of both read 1 and read 2. We performed paired-end sequencing with 300-base reads on an Illumina MiSeq at the University of Texas at Austin Genome Sequencing and Analysis Facility. Reads were demultiplexed using a custom python script. We trimmed barcodes and adapter sequences using *Trimmomatic* version 0.38 (*Bolger et al., 2014*).

When available, we combined short-read data from different platforms before mutation identification. We identified mutations using *breseq* version 0.33.2 (*Deatherage and Barrick, 2014*). We used a bash script called 'generate-LCA.sh' to infer the last common ancestor (LCA) of all evolved strains by taking the intersection of mutations found in previously curated genomes for CZB152 and CZB154; those curated founder genomes (and others) are available at: https://github.com/barricklab/LTEE-Ecoli (*Barrick, 2015*). We further analyzed the mutations called by *breseq* relative to

the LCA using custom python and R scripts available and described at: https://github.com/rohan-maddamsetti/DM0-evolution (*Maddamsetti, 2019*).

We used the following algorithm to find copy-number variation in the genomes. The *breseq* pipeline models $1\times$ copy number using a negative binomial distribution fit to coverage, truncating high and low coverage that might be caused by amplifications and deletions, respectively. We then identified all positions in the genome that rejected that negative binomial at an uncorrected $p$ = 0.05. Finally, we calculated a Bonferroni-corrected $p$-value for contiguous stretches of the genome in which the $1\times$ null model was rejected at each site. We examined coverage at sites separated by the maximum read length to ensure they were not spanned by a single read. For example, in the case of a region of elevated coverage that was 1000 bp in length, covered by 150-base Illumina sequencing reads, the value of $P(coverage = min)^6$ would be calculated, where *min* is the minimum coverage in that region, $P(coverage = min)$ is the probability of that minimum coverage under the negative binomial null model, and six represents the (integer) number of sites that are 150 bp apart in the 1000 bp stretch. The output was then filtered for regions longer than $2 \times 150$ = 300 bp to remove potential false positives. The Bonferroni calculation included corrections for checking every site in the genome in addition to the number of sites that passed the initial 0.05 cutoff for deviations from the negative binomial expectation. All gene amplifications detected in the DM0- and DM25-evolved genomes are reported in *Supplementary file 2*.

## Statistical test for selection on parallel IS*150* insertions

To test for positive selection on parallel IS*150* insertions, we simulated a null model of insertion-site preferences based on the observed data. We conservatively assumed that IS*150* elements can only insert into the positions where we observed insertions in one or more sequenced genomes from either this experiment or the LTEE (*Tenaillon et al., 2016*). We also assumed that the probability of IS*150* transposing into a given site is proportional to the observed number of IS*150* insertions at that site across the sequenced genomes, as would be the case if mutational biases alone accounted for the parallel IS*150* insertions. We then used the non-parametric bootstrap method (100,000 replicates) to calculate the probability that any particular site would be hit by so many IS*150* elements among the DM0-evolved genomes, holding the number of IS insertions over that group fixed.

## RNA-Seq and transcriptome analysis

We performed RNA-Seq on six clones: the three Cit$^+$ clones from the LTEE used as ancestors in our evolution experiment (CZB151, CZB152, and CZB154) and three evolved descendants isolated after 2500 generations of adaptation to DM0 (ZDBp877, ZDBp883, and ZDBp889). We revived each clone from a frozen stock in LB as described above. For preconditioning to minimal medium, we diluted each culture 10,000-fold into DM25 with four-fold replication and allowed them to grow for 24 hr at 37°C with 120 rpm orbital shaking. We then diluted the 16 resulting cultures 100-fold in DM0 and grew them for 48 hr at 37°C with shaking, for preconditioning to the citrate-only medium. We diluted the mature cultures 100-fold again into fresh DM0, and grew them to OD$_{600}$ 0.2 – 0.3, corresponding to mid-log phase, at which point we extracted their RNA using the cold phenol-ethanol method (*Bhagwat et al., 2003*). We recovered RNA using a Qiagen RNeasy MiniKit (#74104), and removed DNA with a Qiagen RNase-free DNase set (#79254). RNA was diluted to 50 ng/mL with nuclease-free water and cDNA amplified by RT-PCR. Purified cDNA was then sequenced by Admera Health (South Plainfield, NJ). We used *kallisto* version 0.44 (*Bray et al., 2016*) to quantify RNA transcripts and *sleuth* (*Pimentel et al., 2017*) to conduct differential-expression analysis and visualization. These results are presented in *Supplementary file 3*.

## Construction of *maeA* plasmid

We constructed a medium-copy-number plasmid based on the kanamycin resistance cassette-containing plasmid, pSB3K3, in which the *maeA* gene was placed under the control of a strong constitutive synthetic promoter and ribosome binding site, P089-R052, described by *Kosuri et al., 2013*. We used PCR to amplify the *maeA* gene from REL606 and the pSB3K3 plasmid. We ordered the P089-R052 promoter as an oligonucleotide. We assembled these components using circular polymerase cloning (*Quan and Tian, 2009*) and Gibson assembly (*Gibson, 2011*). We performed drop dialysis using Millipore membrane filters (VSWP01300) for 15 min to desalt the assembly reactions

before electroporation. We isolated transformants on LB-Kanamycin plates and used PCR to find colonies that contained the P089-R052–*maeA* insert. We used Sanger-sequencing of plasmid inserts to verify that no unintended point mutations had occurred during construction. We designated the final plasmid containing the P089-R052-*maeA* insert in the pSB3K3 backbone RM4.6.2.

## Competition experiments to assess fitness effects of *maeA*

We transformed the Cit$^+$ ancestral clones CZB151 and CZB152 and their Ara$^+$ revertants, ZDB67 and ZDB68, respectively, with the plasmid RM4.6.2. We also transformed the same clones with the empty pSB3K3 vector. We froze stock cultures of each transformant at −80°C with glycerol as a cryoprotectant.

We competed each RM4.6.2 transformant against its cognate pSB3K3 transformant in the clone with the opposite Ara marker state. Briefly, we revived all eight transformants in LB supplemented with 50 μg/mL kanamycin and grew them overnight at 37°C with 120 rpm orbital shaking. We then diluted each overnight culture 10,000-fold in 9.9 mL DM0 and incubated for 48 hr at 37°C with orbital shaking, after which they were diluted 100-fold in fresh DM0 every 48 hr three times to acclimate cells to the citrate-only resource environment. We commenced the competition assays the next day by inoculating 9.9 mL DM0 with 50 μL each of an RM4.6.2 transformant and the oppositely marked pSB3K3 transformant, with fourfold replication for a total of 16 competitions. We ran three-day competitions to estimate fitness as described above.

## Data availability statement

All analysis and statistical scripts have been deposited at www.datadryad.org (https://doi.org/10.5061/dryad.7wm37pvpp). RNA-Seq data have been deposited in the NCBI SRA under accession PRJNA553503. Genome sequencing data have been deposited in the NCBI SRA under accession PRJNA595472. Analysis code is also available at: https://github.com/rohanmaddamsetti/DM0-evolution (*Maddamsetti, 2019*; copy archived at https://github.com/elifesciences-publications/DM0-evolution).

# Acknowledgements

We thank Joshua Franklin and Yann Dufour for helpful discussions and assistance with the microscopy work; Simon D'Alton for assistance with genome sequencing; Daniel Barich for help in handling transcriptomics data; Neerja Hajela and Devin Lake for assistance in the laboratory; Jean Vila, Erik Quandt, Daniel Deatherage, Dacia Leon, Debora Marks, David Ding, Yarden Katz, Helen Murphy, and Kyle Card for helpful discussions; and Sandeep Venkataram, Sébastien Wielgoss, and anonymous reviewers for constructive feedback on previous versions of the manuscript.

# Additional information

## Competing interests

Jeffrey E Barrick: Jeffrey E Barrick is the owner of Evolvomics LLC. The other authors declare that no competing interests exist.

## Funding

| Funder | Grant reference number | Author |
| --- | --- | --- |
| Michigan State University | Ralph Evans Award | Zachary David Blount |
| Kenyon College | Individual Faculty Development Award | Zachary David Blount |
| Michigan State University | Rudolph Hugh Award | Nkrumah A Grant |
| National Science Foundation | DEB-1451740 | Richard E Lenski |
| National Science Foundation | DBI-0939454 | Richard E Lenski |
| USDA National Institute of Food and Agriculture | MICL02253 | Richard E Lenski |

| National Science Foundation | MCB-1923077 | Joan L Slonczewski |

The funders had no role in study design, data collection and interpretation, or the decision to submit the work for publication.

## Author contributions

Zachary D Blount, Conceptualization, Data curation, Formal analysis, Supervision, Investigation, Visualization, Methodology, Writing - original draft, Project administration, Writing - review and editing; Rohan Maddamsetti, Data curation, Software, Formal analysis, Supervision, Investigation, Visualization, Methodology, Writing - original draft, Project administration, Writing - review and editing; Nkrumah A Grant, Data curation, Software, Formal analysis, Investigation, Visualization, Methodology, Writing - original draft, Writing - review and editing; Sumaya T Ahmed, Tanush Jagdish, Jessica A Baxter, Brooke A Sommerfeld, Alice Tillman, Jeremy Moore, Investigation; Joan L Slonczewski, Richard E Lenski, Resources, Supervision, Funding acquisition, Methodology, Project administration, Writing - review and editing; Jeffrey E Barrick, Resources, Data curation, Software, Formal analysis, Supervision, Funding acquisition, Writing - original draft, Project administration, Writing - review and editing

## Author ORCIDs

Zachary D Blount (iD) https://orcid.org/0000-0001-5153-0034
Rohan Maddamsetti (iD) https://orcid.org/0000-0003-3370-092X
Nkrumah A Grant (iD) https://orcid.org/0000-0002-4555-5283
Joan L Slonczewski (iD) http://orcid.org/0000-0003-3484-1564
Jeffrey E Barrick (iD) https://orcid.org/0000-0003-0888-7358
Richard E Lenski (iD) https://orcid.org/0000-0002-1064-8375

## Decision letter and Author response

Decision letter https://doi.org/10.7554/eLife.55414.sa1
Author response https://doi.org/10.7554/eLife.55414.sa2

# Additional files

## Supplementary files

- Supplementary file 1. Key resources table.
- Supplementary file 2. Full results for gene amplifications in the DM0- and DM25-evolved genomes.
- Supplementary file 3. Full results of differential expression analysis between ancestral (CZB151 and CZB152) and evolved strains (ZDBp877, ZDBp883, ZDBp889), calculated using the Wald test implemented in *sleuth* (*Pimentel et al., 2017*).
- Supplementary file 4. Details of populations, clones, and genome sequencing datasets described in this study.
- Supplementary file 5. Details of the mutations found in the DM0- and DM25-evolved clones, as shown in *Figure 11*.
- Transparent reporting form

## Data availability

All analysis and statistical scripts have been deposited at www.datadryad.org (https://doi.org/10.5061/dryad.7wm37pvpp). RNA-Seq data have been deposited in the NCBI SRA under accession PRJNA553503. Genome sequencing data have been deposited in the NCBI SRA under accession PRJNA595472. Analysis code is also available at https://github.com/rohanmaddamsetti/DM0-evolution (copy archived at https://github.com/elifesciences-publications/DM0-evolution).

The following datasets were generated:

| Author(s) | Year | Dataset title | Dataset URL | Database and Identifier |
|---|---|---|---|---|
| Blount ZD, Maddamsetti R, Grant NA, Ahmed ST, Jagdish T, Baxter JA, Sommerfeld BA, Tillman A, Moore J, Slonczewski JL, Barrick JE, Lenski RE | 2020 | Genomic and phenotypic evolution of Escherichia coli in a novel citrate-only resource environment | https://doi.org/10.5061/dryad.7wm37pvpp | Dryad Digital Repository, 10.5061/dryad.7wm37pvpp |
| Blount ZD, Maddamsetti R, Grant NA, Ahmed ST, Jagdish T, Baxter JA, Sommerfeld BA, Tillman A, Moore J, Slonczewski JL, Barrick JE, Lenski RE | 2020 | Genomic and phenotypic evolution of Escherichia coli in a novel citrate-only resource environment | https://www.ncbi.nlm.nih.gov/bioproject/PRJNA595472/ | NCBI BioProject, PRJNA595472 |
| Blount ZD, Maddamsetti R, Grant NA, Ahmed ST, Jagdish T, Baxter JA, Sommerfeld BA, Tillman A, Moore J, Slonczewski JL, Barrick JE, Lenski RE | 2020 | Genomic and phenotypic evolution of Escherichia coli in a novel citrate-only resource environment | https://www.ncbi.nlm.nih.gov/bioproject/PRJNA553503/ | NCBI BioProject, PRJNA553503 |

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
