## [Decision Letter]

**Acceptance summary:**

Understanding the evolutionary challenges faced by organisms as they respond to new environmental conditions is a significant, but often intractable problem. The discovery of *Escherichia coli* lineages from long-term experimental evolution populations that evolved ability to metabolise citrate under aerobic conditions has provided a window to deeper understanding. Of particular note are dynamic mutational processes underpinned by amplifications and insertion sequence movement, and the curious, but entirely believable finding, that adaptation to the new citrate environment is limited by physiological constraints.

**Decision letter after peer review:**

Thank you for submitting your article "Genomic and phenotypic evolution of *Escherichia coli* in a novel citrate-only resource environment" for consideration by *eLife*. Your article has been reviewed by two peer reviewers, and the evaluation has been overseen by a Reviewing Editor and Patricia Wittkopp as the Senior Editor. The reviewers have opted to remain anonymous.

The reviewers have discussed the reviews with one another and the Reviewing Editor has drafted this decision to help you prepare a revised submission.

All reviewers and editors find the work a useful extension of the cit^+^ (invasion of new niche) angle arising from the LTEE and appreciate the new insights particularly concerning structural variant mutations. The work should be broadly valued by the field and especially understanding of how capacity to utilise citrate has become refined during the course of the selection experiment.

Summary:

Blount et al. founded 24 experimental populations from cit^+^ clones, 12 replicates were evolved in DM25, and 12 in DM0. At the end of the evolution experiment the authors characterised growth phenotypes, sequenced clones and followed up with some plasmid complementation and gene expression analysis. The systematic and thorough follow up on the evolved clone from DM0-2, ZDBp874 is a good example of the elegant experimental design and clear writing that characterise this study- it is very easy to follow. Overall, the genetic characterisation of the populations is thorough, although this part would have benefited from temporal analysis of mutations. Arguably the most interesting part of the work arises from phenotypic analysis of derived lines, but here we have concerns regarding the conclusions.

Major concerns:

All reviewers were surprised to find that derived lines were not subject to standard population-level assays of competitive fitness. Instead, OD was used as a phenotypic measure of performance, sometimes with populations, but the more significant data came from analysis of individual clones. Reliance on OD, without calibration against viable cell counts, can result in misleading conclusions. Small differences in death, or cell size, all impact on OD. In a similar vein, Figure 6—figure supplement 1: Some populations (both ancestors and evolved strains) appear to reach a much higher OD. This receives no comment but it would appear to be a significant observation. There also seems to be very large differences between replicates of the evolved clones and ancestors in Figure 7—figure supplement 1/Figure 8—figure supplement 1 that should be commented on in the text as this is quite atypical for growth curves measurements of clones, in that there are more than two-fold differences in endpoint OD.

Of real interest is the evolution of cell death, however, cell death assays are based on only 5 clones and no population-level assay. The five clones include only two clones evolved in this experiment, (including ZDBp910 which is particularly sick in DM 0). Clones do not perform as well as populations in the growth assays, and it is reasonable to expect that clones do not perform as well as populations in the mortality assays. In other words, reliance on growth assays (Figure 3) does not provide knowledge of improvements in the population growth parameters that are evident in the population growth assays (Figure 2). Without carrying out population-based mortality assay, for multiple populations, it is not clear what trait has actually evolved in the population.

Strain ZDBp871 did not show improvement in DM0 and was worse in DM25, yet it was chosen for microscope analysis of cell death. Why were clones that showed large improvements included in the cell death analysis? It would be of major interest for the conclusions of the article to determine if there are strains that evolved reduced cell death. This seems like this is an unnecessary omission that could quite easily be fixed.

One of the claims that follows from the mortality experiments is that there is a mismatch between *E. coli* physiology and the citrate only environment. This is an interesting idea, and the discussion in paragraph four of the Discussion is compelling, but very speculative, with no data to support the idea of pleiotropy, tradeoffs or a relationship between cit^+^ and mortality, aside from the mortality assays in 5 clones.

The mortality assay measures the number of dead cells at stationary phase- cells die at a faster rate in stationary phase than log phase. Since the evolved clones have a reduced lag time, by the time you come to measure mortality these have been in stationary phase for longer than cells that have remained in log phase for a greater proportion of the 24hrs. Could this explain greater number of dead cells in evolved cit0?

It looks like every clone isolated from the LTEE has a high mortality rate- could it be that high mortality is a trait of the LTEE and not specific to cit^+^?

Essential revisions:

If time and resources were unlimited we would request the inclusion of population-based competitive fitness measures of all derived lines relative to the respective ancestor, plus calibration of OD against cell viability, and a more comprehensive set of cell death assays in order to support the claims of increased mortality.

Recognising that both time and resources are limited, plus the fact that there is much that is valuable in the paper, we urge the authors to consider the above requests, but in terms of required revisions, we insist solely on inclusion of additional cell death assays sufficient to bolster claims and address the concerns raised above. We think these can be done quickly.

Even with the addition of further analysis of experimental lines, we think it will likely be necessary to dial back claims about the evolution of increased mortality. We also think that in the absence of population measures of competitive fitness that limitations of the fitness measures should be clearly stated and discussed.

One addition request is to pay careful attention to the degree of repetition. The is opportunity to produce a more streamlined manuscript through removal of extraneous paragraphs and those that are repeated. For example, the Conclusion section is long and largely reads as a second discussion that repeats parts of the first. How about removing repeated sections and producing a succinct conclusion?

---

## [Author Response]

Major concerns:All reviewers were surprised to find that derived lines were not subject to standard population-level assays of competitive fitness. Instead, OD was used as a phenotypic measure of performance, sometimes with populations, but the more significant data came from analysis of individual clones. Reliance on OD, without calibration against viable cell counts, can result in misleading conclusions. Small differences in death, or cell size, all impact on OD.

We have now included results from 1-day competition assays using the evolved populations. Unfortunately, we were unable to conduct longer competitions due to the research shutdown caused by the pandemic. We have also included the results of 3-day competition assays on a subset of evolved clones (for which we isolated neutral Ara^+^/Ara^−^ mutants of their ancestors for comparison). We present these data in the revised text in a new subsection of the Results. We show the fitness data in Figures 3 and 4 of our revised manuscript. Jessica A. Baxter performed part of these competitions. We have in consequence added her as a co-author. The other authors of the manuscript have approved her addition and placement in the author list. We would also emphasize that we do not use OD per se as a measure of performance; rather, we use *rates of change* in OD to estimate some standard growth parameters. The inclusion of both competitive fitness assays and growth-parameter estimates should make the differences in these approaches clear.

In a similar vein, Figure 6—figure supplement 1: Some populations (both ancestors and evolved strains) appear to reach a much higher OD. This receives no comment but it would appear to be a significant observation. There also seems to be very large differences between replicates of the evolved clones and ancestors in Figure 7—figure supplement 1/Figure 8—figure supplement 1 that should be commented on in the text as this is quite atypical for growth curves measurements of clones, in that there are more than two-fold differences in endpoint OD.

Large variations in the OD of replicate cultures of the same clone during growth curve measures are not unusual for Cit^+^ clones, and we reported similar variation in our previous publications. This variation has made it challenging to characterize their growth parameters and fitness. While such extreme variation in OD values is atypical of most study systems, it is entirely consistent with the several lines of evidence we present for a persistent mismatch between *E. coli* physiology and growth on citrate, as it implies the cells are extremely sensitive to subtle environmental fluctuations when growing on citrate. We now discuss these issues thoroughly in the revised text.

Of real interest is the evolution of cell death, however, cell death assays are based on only 5 clones and no population-level assay. The five clones include only two clones evolved in this experiment, (including ZDBp910 which is particularly sick in DM 0). Clones do not perform as well as populations in the growth assays, and it is reasonable to expect that clones do not perform as well as populations in the mortality assays. In other words, reliance on growth assays (Figure 3) does not provide knowledge of improvements in the population growth parameters that are evident in the population growth assays (Figure 2). Without carrying out population-based mortality assay, for multiple populations, it is not clear what trait has actually evolved in the population.

We agree that our analyses of cell death would benefit from greater replication and scope. We planned additional experiments to look more closely at the phenomenon of cell death, but we were unable to conduct them before the COVID-19 pandemic necessitated a research hiatus. We think the fitness data we have included, especially those that show paradoxical fitness declines during evolution, are consistent with the cell-death data we have presented, and we discuss them in that context. Nonetheless, in the absence of follow-up experiments on cell death, we have tempered our claims on this issue. That said, it remains clear that, for at least some combinations of strains and environments, there is substantial cell death, which is indicative of the mismatch between *E. coli* physiology and growth on citrate.

Strain ZDBp871 did not show improvement in DM0 and was worse in DM25, yet it was chosen for microscope analysis of cell death. Why were clones that showed large improvements included in the cell death analysis?

We chose these two evolved strains without considering (and prior to knowing) their growth characteristics. We designed this experiment as a simple demonstration of extensive death of Cit^+^ cells. We were motivated by some preliminary microscopy observations of many ghost-like cells, suggestive of mortality, in the original Cit^+^ LTEE population; those observations were made in the context of a separate (as yet unpublished) study on the evolution of cell size in the LTEE.

Our consideration was simply choosing one DM0-evolved clone (ZDBp871), one DM25-evolved clone, and their common ancestor, CZB151. We chose to use strains descended from CZB151 because the Ara^+^ derivative of that ancestor, ZDB67, has no secondary mutations. The absence of secondary mutations meant we could better relate the cell-mortality results to the fitness analyses of these same strains relative to their ancestor.

It would be of major interest for the conclusions of the article to determine if there are strains that evolved reduced cell death. This seems like this is an unnecessary omission that could quite easily be fixed.

We agree that it would be interesting to find genotypes that evolved reduced death, if they exist. As explained above, we intend to carry out additional experiments on cell mortality in future work, but we are unable to do so at present due to the COVID-19 pandemic.

One of the claims that follows from the mortality experiments is that there is a mismatch between *E. coli* physiology and the citrate only environment. This is an interesting idea, and the discussion in paragraph four of the Discussion is compelling, but very speculative, with no data to support the idea of pleiotropy, tradeoffs or a relationship between cit^+^ and mortality, aside from the mortality assays in 5 clones.

We are pleased that you found this discussion compelling, albeit speculative. Given our limited data on mortality, and the fact that we cannot perform further experiments at this time, we have eliminated the more speculative aspects of the discussion.

That said, our additional fitness results provide yet more evidence to support our hypothesis of a mismatch between *E. coli* cell physiology and the citrate-only environment. The variation among replicate fitness assays, the limited fitness gains even after 2,500 generations, the striking variability across replicate growth curves, and the heightened mortality in the Cit^+^ clones—all indicate that these bacteria struggle to grow normally on citrate. In future work, we will try to more rigorously examine the relationship between the Cit^+^ trait and mortality, and the possible role of pleiotropic tradeoffs.

The mortality assay measures the number of dead cells at stationary phase- cells die at a faster rate in stationary phase than log phase. Since the evolved clones have a reduced lag time, by the time you come to measure mortality these have been in stationary phase for longer than cells that have remained in log phase for a greater proportion of the 24hrs. Could this explain greater number of dead cells in evolved cit0?

Thank you for these ideas. It would be interesting if cell death correlates with time spent in stationary phase (even over the relatively short duration of our growth curve experiments), and if the shorter lag phase could then account for increased death. We hope to answer this question in future experiments (once lab work can be safely resumed) with live/dead staining on samples taken at various stages of population growth. We have revised the text to present the cell-death results that we have for now in as straightforward a way as possible, to avoid possible over-interpretation.

It looks like every clone isolated from the LTEE has a high mortality rate- could it be that high mortality is a trait of the LTEE and not specific to cit^+^?

Similar preliminary and unpublished live-dead staining performed on other LTEE populations indicates that elevated cell death is specific to the Cit^+^ phenotype, with little to no death occurring in Cit^–^strains. This inference is also supported by the observation (now mentioned in our revised paper) of many “ghost-like” cells only in the Cit^+^ population, which first raised the possibility of elevated cell death in this lineage. In addition, Vasi et al., 1994, directly estimated cell mortality (based on viable CFUs) during stationary phase for the ancestor and early-generation samples of all the LTEE populations, and found very little mortaility.

Essential revisions:If time and resources were unlimited we would request the inclusion of population-based competitive fitness measures of all derived lines relative to the respective ancestor, plus calibration of OD against cell viability, and a more comprehensive set of cell death assays in order to support the claims of increased mortality.Recognising that both time and resources are limited, plus the fact that there is much that is valuable in the paper, we urge the authors to consider the above requests, but in terms of required revisions, we insist solely on inclusion of additional cell death assays sufficient to bolster claims and address the concerns raised above. We think these can be done quickly.

We were able to perform some of the recommended fitness assays, but we were unable to perform the requested cell-death experiments due to the closure of the lab caused by the COVID-19 pandemic.

Even with the addition of further analysis of experimental lines, we think it will likely be necessary to dial back claims about the evolution of increased mortality.

We have tempered speculation concerning the evolution of increased cell mortality, presenting the results we have to date as straightforwardly as possible. We provide these data as one of several lines of evidence that cells grown on citrate suffer from substantial physiological stress.

We also think that in the absence of population measures of competitive fitness that limitations of the fitness measures should be clearly stated and discussed.

We believe that the population-level fitness measures we now include satisfy this request.

One addition request is to pay careful attention to the degree of repetition. The is opportunity to produce a more streamlined manuscript through removal of extraneous paragraphs and those that are repeated. For example, the Conclusion section is long and largely reads as a second discussion that repeats parts of the first. How about removing repeated sections and producing a succinct conclusion?

We have revised the manuscript specifically to remove repetition and extraneous paragraphs. In particular, we eliminated the Conclusion as a separate section, retaining some parts where it was more concise than the previous Discussion, largely deleting the preceding text, and editing the structure and organization. Thank you for this advice; we believe our revised paper is both shorter and clearer.